

# Comparison of ground-based and satellite measurements of water vapour vertical profiles over Ellesmere Island, Nunavut

Dan Weaver[1], Kimberly Strong[1], Kaley A. Walker[1], Chris Sioris[2], Matthias Schneider[3], C. Thomas McElroy[4], Holger Vömel[5], Michael Sommer[6], Katja Weigel[7], Alexei Rozanov[7], John P. Burrows[7], William G. Read[8], Evan Fishbein[8], and Gabriele Stiller[3]

[1]Department of Physics, University of Toronto, Toronto, Ontario, Canada

[2]Environment and Climate Change Canada, Toronto, Ontario, Canada

[3]Institute of Meteorology and Climate Research (IMK-ASF), Karlsruhe Institute of Technology, Karlsruhe, Germany

[4]Department of Earth and Space Science and Engineering, York University, Toronto, Canada

[5]Earth Observing Laboratory, NCAR, Boulder, Colorado, USA

[6]GRUAN Lead Centre, Deutscher Wetterdienst, Lindenberg, Germany

[7]Institute of Environmental Physics, University of Bremen, Bremen, Germany.

[8]Jet Propulsion Laboratory, California Institute of Technology, Pasadena, California, USA

*Correspondence to:* Dan Weaver (dweaver@atmosp.physics.utoronto.ca)

**Abstract.** Improving measurements of water vapour in the lower stratosphere and upper troposphere (UTLS) is a priority for the atmospheric science community. In this work, UTLS water vapour profiles derived from Atmospheric Chemistry Experiment (ACE) satellite measurements are assessed with coincident ground-based measurements taken at a high Arctic

observatory at Eureka, Nunavut, Canada. Additional comparisons to satellite measurements taken by AIRS, MIPAS, MLS, SCIAMACHY, and TES are included to put the ACE-FTS and ACE-MAESTRO results in context.

Measurements of water vapour profiles at Eureka are made using a Bruker 125HR solar absorption Fourier transform infrared spectrometer at the Polar Environment Atmospheric Research Laboratory (PEARL) and radiosondes launched from the Eureka Weather Station. Radiosonde measurements used in this study have been processed with software developed by the Global

Climate Observing System (GCOS) Reference Upper Air Network (GRUAN) to account for known biases and calculate uncertainties in a well-documented and consistent manner.

ACE-FTS measurements were within 11 ppmv (13%) of 125HR measurements between 6 and 14 km. Between 8 and 14 km ACE-FTS profiles showed a small wet bias of approximately 8% relative to the 125HR. ACE-FTS water vapour profiles had mean differences of 13 ppmv (32%) or better when compared to coincident radiosonde profiles at altitudes between 6 and

14 km; mean differences were within 6 ppmv (12%) between 7 and 11 km. ACE-MAESTRO profiles showed a small dry bias relative to the 125HR of approximately 7% between 6 and 9 km and 10% between 10 and 14 km. ACE-MAESTRO profiles agreed within 30 ppmv (36%) of the radiosondes between 7 and 14 km. ACE-FTS and ACE-MAESTRO comparison results





show closer agreement with the radiosondes and PEARL 125HR overall than other satellite datasets - except AIRS. Close agreement was observed between AIRS and the 125HR and radiosonde measurements, with mean differences within 5% and correlation coefficients above 0.83 in the troposphere between 1 and 7 km.

Comparisons to MLS at altitudes around 10 km showed a dry bias, e.g., mean differences between MLS and radiosondes were -25.6%. SCIAMACHY comparisons were very limited due to minimal overlap between the vertical extent of the measurements. TES had no temporal overlap with the radiosonde dataset used in this study. Comparisons between TES and the 125HR showed a wet bias of approximately 25% in the UTLS and mean differences within 14% below 5 km.

## 1.  Introduction

Atmospheric water vapour plays a crucial role in the chemistry, dynamics, and radiative balance of the Earth's atmosphere. In the upper troposphere and lower stratosphere (UTLS), water vapour has significant impacts on radiative forcing (Solomon et al. 2010; Soden et al., 2008). Changes in UTLS water vapour abundances have important implications for climate (Dessler and Sherwood, 2009; Held and Soden, 2000). Indeed, the Global Climate Observing System (GCOS) considers acquiring measurements of water vapour profiles to an accuracy of 5% essential for understanding the climate system (GCOS, 2016). However, global measurements of UTLS water vapour are not yet acquired routinely at the accuracy sought by the atmospheric science community. Instruments and measurement techniques are being developed to fill this observational need.

Ground-based observations of water vapour are made using a variety of instruments. Many instruments only acquire total column measurements, e.g., Sun photometers. Others acquire profiles as well, such as Fourier transform infrared (FTIR) spectrometers. However, ground-based FTIR observations are limited by the relatively sparse network of sites globally, and current FTIR water vapour profile retrievals have a modest vertical resolution (e.g., Schneider et al., 2016). Balloon-based radiosonde sensors measure atmospheric humidity profiles with high vertical resolution, typically better than 100 m, and are launched daily from approximately 1000 sites globally (Durre et al., 2006). This geographic coverage nonetheless has many gaps, e.g. in the polar and oceanic regions, and radiosonde launches are typically limited to once or twice a day. While limited in their global coverage, ground-based instruments produce well-characterized measurements that can be used to study specific sites, compare with models, and validate satellite measurements.

Satellite-based measurements complement ground-based observations by producing frequent global measurements of atmospheric constituents. More than a dozen satellites are currently (or have been recently) making measurements of water vapour. There is interest in assessing the accuracy and quality of these datasets. For example, a World Climate Research Programme (WCRP) Stratosphere-troposphere Processes And their Role in Climate (SPARC) activity is currently conducting a comprehensive overview of stratospheric and lower mesospheric water vapour satellite measurements. This effort, the second SPARC water vapour assessment (WAVAS-II), intercompares the available satellite measurements to understand the





differences between available datasets, measurement uncertainties, and the trends in stratospheric and lower mesospheric water vapour. Results from the WAVAS-II effort are being published in a special inter-journal issue of AMT/ACP/ESSD, e.g., Khosrawi et al. (2018), and is available at: https://www.atmos-chem-phys.net/special_issue830.html.

Developing highly accurate and vertically-resolved UTLS water vapour profile measurements from satellite instruments is a
priority of the atmospheric observing community (Müller et al., 2015). However, obtaining sensitivity to the troposphere and producing high vertical resolution profiles is challenging for many satellite instruments. Comparisons to ground-based observations offer an opportunity to assess the accuracy of satellite measurements.

The objective of this study is to assess the Arctic water vapour profiles retrieved from Atmospheric Chemistry Experiment (ACE) satellite observations using comparisons to coincident measurements taken at a Canadian high Arctic observatory in
Eureka, Nunavut. In addition, other satellite instruments with Eureka-coincident water vapour profile measurements are compared to put the ACE results in the context of the broader effort to measure water vapour from satellites. This study adds to earlier work that has compared ground-based FTIR measurements to ACE v3.5/3.6 (e.g., Griffin et al., 2017) and studies comparing ACE measurements to those of other satellites (e.g., Sheese et al., 2017). This study is structured as follows. Section 1 introduces the motivation for UTLS water vapour measurements and describes the ground-based measurement site. Section
2 describes the instruments and datasets used in the study. Section 3 compares the satellite and ground-based measurements, noting the methods used to match observations and account for different vertical sensitivities. Section 4 discusses the results of the comparisons. Section 5 offers conclusions about the ability of the ACE and other satellite datasets to contribute to our knowledge of high Arctic water vapour and comments on the implications for future research.

### 1.1.  Ground-based reference site

Eureka, Nunavut is a research site on Ellesmere Island in the Canadian high Arctic. It has an extremely cold and dry environment. Eureka is located at 10 metres above sea level on the shore of Slidre Fjord, 12 km east of Eureka Sound. Open water occurs regionally during summer, but during the rest of the year, the surface of the fjords and sounds are frozen. The geography of the surrounding area is variable, including ridges, hills, and small mountains. Because of the site's 80° N latitude, there is no sunlight between mid-October and mid-February.

Environment and Climate Change Canada's Eureka Weather Station (EWS) is the primary presence in Eureka (79.98° N, 85.93° W). One of the key measurements taken at the EWS is the twice-daily radiosonde observations of temperature, pressure, wind, and humidity profiles. Radiosonde measurements at Eureka extend back to 1948. These measurements show, for example, that tropospheric temperatures are increasing, water vapour total columns are increasing, and temperature and humidity inversions often form in the lower troposphere above Eureka between fall and spring
(Lesins et al., 2010). The EWS is also used as an operational hub for government and academic research conducted in the area. Since 2006, the Canadian Network for the Detection of Atmospheric Change (CANDAC) has operated a large suite of atmospheric monitoring instruments at the Polar Environment Atmospheric Research Laboratory (PEARL) near Eureka (Fogal et al., 2013). The Ridge Lab is the largest of the PEARL facilities, and is located at 80.05° N, 86.4° W on top of a ridge at



610 m elevation, 10 km west of Eureka. The large number of observations taken by the EWS and PEARL instruments offer extensive characterization of atmospheric conditions at the site.

Many polar-orbiting and limb-viewing satellites commonly have overpasses with Eureka. As a result, measurements taken at PEARL have contributed to many validation studies, e.g. of ACE (Griffin et al., 2017), MOPITT (Buchholz et al., 2017),
OCO-2 (Wunch et al., 2017), and OSIRIS (Adams et al. 2012).

## 2. Instruments

This section presents the water vapour datasets from Eureka ground-based instruments and Eureka-coincident satellite instruments that are used in this study. Table 1 summarizes the available datasets, notes the technique, retrieval version, and how often measurements are taken. Figure 1 illustrates the temporal availability of atmospheric water vapour measurement
from each instrument. Figure 2 illustrates the vertical ranges of the datasets.

### 2.1. Radiosondes

Radiosondes are launched by the EWS twice a day (11:15 and 23:15 UT) using hydrogen-filled balloons. Occasionally, additional radiosondes are launched at other times of day for campaigns. The balloons typically reach the middle of the stratosphere (i.e., 30-33 km) before bursting.

The EWS used Vaisala-built RS92 radiosonde models during the timeframe examined in this study. These sensors are widely used by meteorological stations around the world. RS92 relative humidity (RH) measurements are made using thin-film capacitance sensors. The variable of interest for this study is volume mixing ratio (VMR) in parts per million by volume (ppmv). RH measurements from the radiosondes can be converted to mixing ratio using:

$$\text{VMR(z)} = \frac{RH(z)\, e_S(T)}{P(z)},$$  (1)

where RH is the relative humidity, P is the pressure at a given altitude (z), and $e_s$ is the temperature-dependent saturation vapour pressure of water vapour with respect to liquid water. The $e_s$ equation of Hyland & Wexler (1983) is used for consistency with Vaisala humidity measurement calibration (Miloshevich, 2006).

As the balloon rises through the atmosphere, there comes a point where the humidity sensor can no longer report a meaningful value. Limiting the radiosonde humidity measurements to below the tropopause height (TPH) or a typical tropopause value
usually ensures that only physically meaningful observations are used; however, this potentially removes valid and useful information.

Eureka radiosonde humidity profiles often have clear structure and information about water vapour above the tropopause, which is typically between 8 and 12 km. Miloshevich et al. (2009) found that the tropopause is not a limiting factor for RS92 humidity measurements, and reported close agreement between bias-corrected radiosonde and frostpoint hygrometer (FPH)





profiles at temperatures below -70˚C and below mixing ratios of 5 ppmv. They recommended limiting radiosondes to pressures greater than 100 hPa during daytime and 75 hPa at night. The mean altitude at which the atmosphere above Eureka has a pressure of 100 hPa is 16.01 km ($\sigma$ = 0.47 km), based on radiosonde measurements between 1961 and 2017. We limit radiosonde humidity measurements to altitudes below 15 km for this study as a quality control measure.

RS92 humidity measurements are also known to be affected by solar heating and low temperature calibration error dry biases, as well as errors due to sensor response lag (Vömel et al., 2007a; Miloshevich et al., 2009). The dry bias caused by solar heating of the sensor is not significant in Eureka during winter due to the lack of sunlight; however, it can affect measurements during the sun-lit portion of the year. The calibration error and time-lag error affect low temperature measurements, and are relevant for Eureka conditions. To correct for known biases in a consistent, transparent, and well-documented manner, Eureka

radiosonde measurements have been processed with software developed by the GRUAN, described by Dirksen et al. (2014). Eureka is not a formal GRUAN-participating site and the data are not a formal GRUAN data product; however, available raw Eureka radiosonde measurement files were processed by the GRUAN team for use in this study. This processing also calculates uncertainties for reported values and recovers flight details (e.g., latitude, longitude). Only raw files between September 3, 2008 and October 7, 2017 were available for processing. Minor gaps within that timeframe exist. In total, 5515 radiosonde

profiles which have been processing using GRUAN methodologies are available for Eureka. They have been quality control-filtered to remove any profile with 'rejected' status.

In the troposphere (and sometimes parts of the lower stratosphere), the uncertainty of Eureka radiosonde water vapour mixing ratio profiles are typically 3 to 5%. Uncertainty in the water vapour mixing ratio, calculated by propagating uncertainties in Eq.1 by quadrature, is dominated by the relative humidity uncertainty. Temperature measurement uncertainties are typically a

few tenths of a degree. Pressures similarly have uncertainties on the order of tenths of a hPa. There are occasionally thin dry layers in the middle troposphere that have larger humidity uncertainty. These profile elements are kept. If there are sections of the profile larger than 500 m in the troposphere with high uncertainty values, the entire profile is filtered out.

In the lower stratosphere, the profile reaches a point where the uncertainty increases rapidly. This point changes from profile-to-profile. We limit each individual water vapour profile to the altitude where this rapid increase in uncertainty occurs by

finding where the uncertainty first reaches 20%. This is typically a few kilometres above the tropopause. Thus, each radiosonde profile has a different altitude range, depending on the height reached by the balloon and the uncertainty of the measurements. The mean altitude reached by the filtered profiles is 11.3 km ($\sigma$ = 4.4 km).

Once launched, radiosonde balloons drift away from the site due to winds. The radiosondes used in this study stayed within a mean distance of 29.8 km ($\sigma$ = 16.5 km) from Eureka while under 15 km altitude. The mean time to reach 15 km altitude was

54.4 minutes ($\sigma$ = 6.2 minutes).



## 2.2. PEARL 125HR

The Bruker-made IFS 125HR FTIR spectrometer used for this study is located at the Ridge Lab. Installed in July 2006, the 125HR records high-resolution (0.0035 cm$^{-1}$) mid-infrared (MIR) solar absorption spectra in the framework of the Network for the Detection of Atmospheric Composition Change (NDACC) (Batchelor et al., 2009). Because this technique relies on

sunlight, measurements require clear-sky conditions. Due to PEARL's 80° N latitude, there are no 125HR measurements from mid-October to mid-February (i.e., during Polar Night). Even in mid-summer, the high latitude FTIR spectrometer measurements occur at relatively large solar zenith angle (SZA). The minimum SZA at Eureka is 56.5°. This measurement geometry means that the 125HR typically samples the atmosphere south of the Ridge Lab. During the 24-hour sunlight of Polar Day, during the high Arctic summer, the Sun's position is north of the instrument during what is usually night. However,

125HR measurements are not made overnight due to on-site operator limitations and the lack of an automated shut down trigger in the case of problematic weather.

The 125HR water vapour dataset used in this study was produced using the retrieval technique summarized in Schneider et al. (2016) and Barthlott et al. (2016), as part of the MUlti-platform remote Sensing of Isotopologues for investigating the Cycle of Atmospheric water (MUSICA) project. MUSICA uses the existing NDACC FTIR spectrometer observations to produce

precise and accurate measurement of water vapour isotopologues. This process applies an Optimal Estimation technique based on Rodgers (2000) and the PROFITT retrieval code of Hase et al. (2004) using a combination of strong and weak absorption features on a logarithmic scale. The accuracy of the MUSICA water vapour profiles is about 10% (Schneider et al., 2016). The sensitivity of the retrieval to the atmosphere (i.e., the sum of the averaging kernel rows) varies seasonally due to the dependence on the SZA. The retrieval is typically sensitive throughout the troposphere (i.e., sensitivity above 0.9) and there is some

sensitivity in the lower stratosphere (e.g., sensitivity above 0.5). The MUSICA retrieval's sensitivity to the lower stratosphere is maximum during March, which is also when ACE coincidences occur with Eureka.

MUSICA ground-based FTIR products nominally exclude measurements recorded at SZAs greater than 78.5°. This filter has been removed for this study. Due to Eureka's high-latitude location, this filter removes all measurements between February and the end of March, as well as between September and mid-October. A study of the MUSICA water vapour total column

dataset derived from the PEARL 125HR showed that the SZA limit was likely unnecessarily strict, as agreement did not change between the 125HR and other instruments when the SZA limit was relaxed (Weaver et al., 2017). Standard quality control of the MUSICA dataset, which was applied to the data used here, is described in detail by Barthlott et al. (2016).

## 2.3. ACE on SCISAT

The Canadian Space Agency's (CSA's) SCISAT was launched into a high-inclination (74°) 650 km altitude Earth orbit on

August 12, 2003. This orbit enables limb-viewing measurements over the polar regions, as well as other latitudes. There are two primary Atmospheric Chemistry Experiment (ACE) instruments aboard SCISAT, ACE-Fourier transform spectrometer (ACE-FTS) and ACE-Measurement of Aerosol Extinction in the Stratosphere and Troposphere Retrieved by Occultation





(ACE-MAESTRO). They share a sun-tracker. ACE solar occultation limb-viewing observations involve keeping the sun-tracker pointed at the Sun as the satellite approaches a sunrise or sunset during its orbit and taking sequences of atmospheric and exo-atmospheric absorption spectra.

Coincidences between ACE and Eureka occur during the months of February, March, September, and October. 348 out of 551
coincidences between ACE and Eureka between August 2006 and March 2017 occurred during February and March.

### 2.3.1. ACE-FTS

ACE-FTS is an FTIR spectrometer built by ABB Inc. It acquires spectra between 750 and 4400 $cm^{-1}$ at a resolution of 0.02 $cm^{-1}$. This series of measurements, taken every 2 seconds, is used to retrieve trace gas profiles between the mid-troposphere and 150 km with a vertical resolution ranging between 3 and 4 km (Boone et al., 2005; 2013). This technique has
a horizontal resolution of ~300 km (Bernath, 2017).

This study uses ACE-FTS v3.6 data, provided on the 1-km altitude grid in water vapour mixing ratio. Measurements with quality control flags identifying outliers, high percent errors, or instrument/processing errors were filtered out, following recommendations in Sheese et al. (2015). The water vapour retrieval is limited to altitudes between 5 and 100 km.

The validation of an earlier version (v2.2) of ACE-FTS (and to a limited extent, ACE-MAESTRO) water vapour retrievals
was examined by Carleer et al. (2008). They concluded that ACE-FTS measurements provide accurate $H_2O$ measurements in the stratosphere (better than 5% from 15–70 km) but expressed no firm conclusions about its water vapour measurements in the upper troposphere. Comparisons to FPH measurements showed a possible small dry bias in ACE-FTS measurements at altitudes near 10 km.

Sheese et al. (2017) examined the current ACE-FTS v3.6 $H_2O$ product (as well as other molecules) by comparing it with co-
located MIPAS and MLS measurements by hemisphere. Correlations between ACE-FTS and MLS were observed to be greater than between ACE-FTS and MIPAS. Their analysis examined stratospheric altitudes, where a mean relative difference in the ACE-FTS water vapour product was observed above 16 km ranging from −12 to 2%. In addition, tight coincidence criteria of 15 minutes and 25 km were applied to examine agreement near the hygropause. A mean dry bias of 20% was observed in ACE-FTS profiles relative to MIPAS v5 and MLS v3.3/3.4 at 13 km altitude.

### 2.3.2. ACE-MAESTRO

ACE-MAESTRO is a dual spectrometer with a wavelength range of 285–1015 nm and a resolution of 1.5 - 2.5 nm (McElroy et al., 2007). The ACE-MAESTRO water vapour retrieval algorithm is described by Sioris et al. (2010) with updates described in Sioris et al. (2016a). Water vapour profiles are retrieved from ACE-MAESTRO optical depth spectra. The tangent height registration of the optical depth spectra relies on matching simulated $O_2$ slant columns obtained from air
density profiles, based on ACE-FTS temperature and pressure, with slant columns observed by ACE-MAESTRO using the $O_2$



A band. The water vapour profiles are retrieved on an altitude grid that matches the vertical sampling. Within 500 km of Eureka, ACE-MAESTRO water vapour profiles include altitudes ranging between 4 and 25 km.

The ACE-MAESTRO dataset is sparser than the ACE-FTS dataset for two main reasons. ACE-MAESTRO pointing determination requires the existence of ACE-FTS data, so the available ACE-MAESTRO occultation events are a subset of the ACE-FTS occultations. In addition, ACE-MAESTRO ozone is a necessary input to the ACE-MAESTRO water vapour retrieval. The ACE-MAESTRO ozone retrieval fails occasionally, causing most of the measurements missing from the ACE-MAESTRO water vapour product relative to ACE-FTS product.

## 2.4. Aqua

The U.S. National Aeronautic and Space Administration (NASA) launched the Aqua satellite into a 705 km altitude Sun-synchronous near-polar orbit on May 4, 2002. Aqua's orbit has a 1:30 pm equatorial crossing time and an inclination of 98.2°. It is part of the A-Train constellation of Earth observation satellites. The primary mission of Aqua instruments is to study the atmospheric component of the global water cycle (Parkinson, 2003).

### 2.4.1. AIRS

The AIRS instrument is a hyperspectral thermal infrared grating spectrometer on board Aqua. Its detector observes Earth-emitted radiance from a nadir-orientation using 2378 channels between 3.7 and 15.7 μm. AIRS acquires an enormous number of measurements, collecting about three million spectra per day (Chahine et al., 2006).

AIRS water vapour retrievals have been used to study processes such as the water vapour feedback (Dessler et al., 2008), to evaluate climate models (Pierce et al., 2006), and to improve numerical weather forecasting (Chahine et al., 2006). AIRS aims to produce dense global measurements of temperature and humidity at an accuracy comparable to radiosondes. This study uses level 2 AIRS retrieval v6 data, described in detail by Susskind et al. (2003, 2014). The standard temperature product contains 28 pressure levels, while the standard water vapour product has 15 pressure levels from 1100 to 50 hPa (e.g., between the surface and approximately 20 km in altitude near Eureka).

Only altitudes that meet the "best" level of quality are used for this study, following the guidelines in the AIRS v6 user guide (Olsen et al., 2017). The altitude range for which AIRS profiles are available varies significantly, with fewer passing the quality control filter at low-tropospheric altitudes. The AIRS retrieval is insensitive to water vapour layers with less than 0.01 mm of integrated water vapour. This approximately translates to water vapour abundances less than 15 ppmv (Olsen et al., 2017), typically affecting profile elements above 15 km near Eureka. AIRS is also limited to altitudes with pressures greater than 100 hPa, and has diminishing sensitivity at altitudes with pressures less than 300 hPa (approximately 9 km near Eureka) (Olsen et al., 2017). As mentioned in the discussion of the radiosondes' altitude range, 100 hPa occurs at approximately 16 km in altitude above Eureka. The relative abundance of AIRS profiles ensures measurements are nonetheless available for comparisons.





## 2.5. Aura

NASA's Aura satellite was launched into a near-polar Sun-synchronous 705 km orbit on July 15, 2004. It is part of the A-train constellation of Earth observing satellites, orbiting 15 minutes behind Aqua. Aura's orbit has a 98.2° inclination and an equatorial crossing time near 1:45 pm local solar time. Instruments aboard Aura, such as the Microwave Limb Sounder (MLS)

and TES, study atmospheric chemistry and dynamics.

### 2.5.1. MLS

MLS measures radiation emitted from the atmosphere from a limb-viewing geometry. The atmosphere is scanned twice each minute as the satellite progresses through an orbit that offers a nearly global coverage, between 82° N and 82° S. MLS measurements have been used to assess ACE as well as other satellite measurements, e.g., Hegglin et al. (2013) and

Sheese et al. (2017). This study uses MLS v4.2 data.

MLS water vapour profiles are vertically resolved at pressures less than 383 hPa. At Eureka, MLS's lower altitude limit of 316 hPa corresponds to altitudes near 8 km. MLS water vapour profiles agree within 1% of FPH measurements in the stratosphere, i.e. at $P < 100$ hPa (Hurst et al., 2014). Hurst et al. (2016) showed that agreement between MLS v4.2 and the FPH measurements began to diverge in 2010 at a rate of approximately 1% per year. At 215 hPa and 316 hPa, MLS v1.5 was

observed to have a dry bias of 11 to 23% relative to 10 geographically dispersed FPH measurement sites (Vömel, 2007b).

### 2.5.2. TES

TES is an FTS aboard Aura that observes emitted radiance between 650 and 3050 cm$^{-1}$ spectral resolution of 0.10 cm$^{-1}$ when observing in nadir mode and 0.025 cm$^{-1}$ limb viewing mode (Beer et al., 2001). Limb scanning measurements were performed only until May 2005. The TES water vapour retrieval uses nadir observations, which have a footprint of 5 km by 8 km. Routine

measurements involve a series of observations continuously for 16 orbits (26 hours).

Measurements are only available near Eureka's high Arctic latitude until September 2008. The latitudinal range of TES measurements was limited to latitudes between 50° S and 70° N in summer 2008 to conserve instrument life (Herman and Osterman, 2014). Measurements were further limited to between 30° S and 50° N in spring 2010. However, high latitude measurements were taken in July 2011 as part of a special observation set.

TES retrieval v6 is used for this study. It is based on an optimal estimation non-linear least-squares approach described by Bowman et al. (2006). The vertical information content of TES profiles varies; retrievals with less than 3 degrees of freedom for signal (DOFS) are filtered out. In the subset of measurements examined in this study, TES DOFS range between 3.0 and 5.2.

Comparisons between TES v5 water vapour and global radiosonde measurements have shown a wet bias of 15% in the middle

troposphere (Herman and Kulawik, 2013). Shephard et al. (2008) compared TES water vapour v3 with radiosondes, finding a wet bias in TES retrievals of between 5% in the lower troposphere and 15% in the upper troposphere.



## 2.6. EnviSat

The European Space Agency (ESA)'s Environmental Satellite (EnviSat) was a large platform for Earth observation instruments. Launched into a polar orbit on March 1, 2002, with an inclination of 98.5° and an equatorial crossing time of 10:00 am mean local solar time. Observations from its ten instruments ended in April 2012. On board were two atmospheric

limb sounders, the Michelson Interferometer for Passive Atmospheric Sounding (MIPAS) and the Scanning Imaging Absorption Spectrometer for Atmospheric Cartography (SCIAMACHY). The decade of measurements taken by MIPAS and SCIAMACHY have been widely used to study atmospheric composition, and are often used in comparisons to other limb sounders.

### 2.6.1. MIPAS

MIPAS is an FTIR spectrometer that observes mid-infrared atmospheric emission from a limb-viewing geometry (Fischer et al., 2008). The spectral resolution of MIPAS was reduced from 0.025 cm$^{-1}$ to 0.0625 cm$^{-1}$ in 2004 due to technical problems. The timeframe examined in this study, 2006 onwards, is entirely during the reduced spectral resolution period. This measurement mode has improved spatial resolution. In polar regions, the nominal tangent altitude spacing is 1.5 km in the UTLS region.

This study uses MIPAS retrieval v5 and v7 from the Institute of Meteorology and Climate Research (IMK). Both retrieval versions cover the same temporal range. This retrieval technique is described by von Clarmann et al. (2009) and uses Tikhonov regularization. In the UTLS, the profiles are provided on a 1-km grid. At 10 km, the vertical resolution (v5) is 3.3 km the horizontal resolution is estimated to be 206 km (von Clarmann et al., 2009). Quality control filtering is applied according to recommended values. MIPAS water vapour data is recommended for use only above 12 km altitude. However, in this study

all available altitudes provided in the official data release are used. MIPAS water vapour profile retrievals reach altitudes as low as 5 km.

Stiller et al. (2012) compared an earlier version of the MIPAS IMK retrieval (v4) with cryogenic frostpoint hygrometer (CFH) measurements of water vapour profiles during the Measurements of Humidity in the Atmosphere and Validation Experiments (MOHAVE) campaign near Pasadena, California in October 2009. Above 12 km, MIPAS showed agreement within 10%.

Results suggest MIPAS v4 water vapour might be 20-40% wet biased around 10 km.

### 2.6.2. SCIAMACHY

SCIAMACHY is an imaging spectrometer that has limb, nadir, and occultation viewing modes (Bovensmann et al., 1999). Limb measurements of scattered sunlight are the basis for the Institut für Umweltphysik (IUP) v3.01 and v4.2 water vapour retrievals used in this study. Both retrieval versions cover the same temporal range. It is based on the optimal-estimation

approach described by Rodgers (2000) using a first-order Tikhonov constraint. The vertical resolution is approximately 3 km. The retrieval calculates a scaling factor for the tropospheric water vapour profile; altitudes below 10 km are not recommended





for use and are not used here. The details of this retrieval are described in Weigel et al. (2016) for v3.01. For v4.2 several changes were implemented first of all to improve the aerosol correction and the vertical resolution. Additionally, v4.2 uses all appropriate SCIAMACHY measurements, v3.01 only a subset. One issue for limb sensing is the number of cloud free scenes. This is limited by the sampling approach, which was constrained by the data rate available on Envisat.

Weigel et al. (2016) compared MIPAS v3.01 to MIPAS v5, MLS v3.3, and other satellite datasets, in 30° latitudinal bands. Results showed SCIAMACHY limb measurements between 10 and 25 km in altitude were reliable between 11 and 23 km, and accurate to about 10% between 14 and 20 km. Below 14 km, differences with other datasets increase to up to 50%, showing a possible SCIAMACHY v3.01 wet bias, which is most pronounced in the tropics and least in the polar latitudes.

## 3. Comparison of water vapour measurements

Water vapour profiles from ACE-FTS, ACE-MAESTRO, AIRS, MIPAS, MLS, SCIAMACHY, and TES were compared with Eureka radiosonde and PEARL 125HR measurements following the methodology described below.

### 3.1. Method

Coincident profile measurements have been compared using difference and correlation plots. Absolute differences and percent relative differences are calculated using:

$difference = X - Y,$ (2)

$\% \, difference = \frac{(X-Y)}{Y} \times 100\%,$ (3)

where $X$ is the satellite measurement and $Y$ is the reference measurement, e.g., 125HR or radiosondes.

To show the overall agreement observed between the measurements, the means of coincident profile differences are calculated. Altitude ranges for which there are measurements available vary for each contributing matched pair of profiles, resulting in a 20 variable number of profiles contributing to comparisons at each altitude. The number of contributing matches at each altitude level is reported in the comparison figures.

In addition to showing profile comparisons, comparisons at specific representative altitudes are presented. These illustrate the extent of the variability in the overall mean agreement between the datasets.

A minimum number of 15 coincidences is needed for results to be reported and shown in the tables and figures. This aimed to 25 balance the reality that there are limited number of coincidences available and the need to ensure there are a meaningful number of comparison results available at each altitude.





### 3.1.1. Coincidence criteria

A three-hour temporal coincidence criterion was used for all comparisons and applied in two ways. Firstly, if multiple coincidences were found within this interval, only the closest pair was kept. Each pair of coincident measurements is thus independent of others contributing to the overall assessment of different measurement techniques. This method often results in a smaller time difference between measurements than is otherwise permitted by the criterion. The comparisons were also performed using all possible coincidences within this criterion. While increasing the number of matches, in some cases significantly, the observed agreement between instruments was similar to that for the first method, which is summarized in Table 2 and Table 3. Results using the first method are discussed below. Results of comparisons where all possible coincidence pairs are used are available in Supplementary Tables 1 and 2.

A 500 km spatial coincidence criterion was also applied. The spatial criterion is similar in scale to the horizontal area covered by a limb-viewing satellite measurement. When calculating the distance between PEARL and an ACE observation, the 30 km (calculated geometrically) tangent height of the ACE measurement was used as the satellite measurement's position. This approach has been used for validation, e.g., Fraser et al., 2008.

The difference in measurement geometries, and the long path of a limb-viewing measurement in particular, can result in ACE-FTS measuring a different airmass than the 125HR and radiosondes. Figure 3 illustrates the variation of water vapour abundances in the region around Eureka using binned AIRS measurements at 400 hPa (corresponding to altitudes between 6.1 and 7.5 km, with a mean altitude of 6.7 km and a standard deviation of 0.2 km) for two sample months, March and July. Variability in the water vapour abundances in the region around Eureka is seen to be larger in the summer than in the winter. October resembles the results shown for March.

### 3.1.2. Smoothing

When comparing satellite profiles with the PEARL 125HR, the comparison instrument's profile was smoothed by the MUSICA averaging kernel of the 125HR measurement to account for the vertical resolution differences between the instruments. The procedure for smoothing followed Rodgers and Connor (2003):

$$x_{smoothed} = A(x - x_a) + x_a, \tag{4}$$

where $x_a$ is the MUSICA a priori profile, $x$ is the comparison instrument profile, and $A$ is the averaging kernel matrix. Since the MUSICA water vapour retrievals are performed on a logarithmic scale, the smoothed profile is calculated using:

$$x_{smoothed} = e^{A(x - x_a) + x_a}, \tag{5}$$

where $x$, $x_a$, and $A$ are in $\log_e$ space.





Before smoothing, the satellite profile was interpolated to the MUSICA retrieval grid and the MUSICA a priori profile was used to fill gaps in the comparison profile (e.g., altitudes beneath the lower limit of satellite measurements). After smoothing, altitudes for which there were no original data were removed.

When comparing satellite measurements to the radiosonde profiles, radiosonde profiles were smoothed using the satellite's averaging kernels where possible, i.e., for SCIAMACHY and TES, following the same procedure described for the 125HR. MIPAS retrievals do not use an a priori profile, so the smoothed radiosonde profile is calculated using:

$$x_{smoothed} = e^{Ax}. \tag{6}$$

In the cases of ACE-FTS, ACE-MAESTRO, AIRS, and MLS, the radiosonde profiles have been smoothed using Gaussian weighting functions with a full width half maximum (FWHM) that approximates the vertical resolution of the satellite measurement. This procedure is used because ACE instruments do not have averaging kernels. MLS has an averaging kernel for use in the polar regions; however, the user's guide states that the use of the water vapour averaging kernel at the lowest valid altitude levels (i.e., lower stratosphere at 316 hPa and 262 hPa) is not recommended (Livesey et al., 2016). Since these altitudes are of particular interest to this study, the MLS averaging kernels are not suitable. AIRS also has averaging kernels, distributed in supplementary data files; however, the AIRS averaging kernels only capture the information added during the final physical retrieval, but not information extracted from the AIRS radiances during the neural network step. The AIRS retrieval uses many channels effectively synthesizing a narrow weighting function, then is possible from any one channels. We use of the width of the AIRS weighting functions to estimate a Gaussian smoother generally overestimates the amount of smoothing. Thus, weighting functions are used in these cases as a reasonable approximate method of smoothing the vertical resolution of these profiles.

To create weighting functions, first, Gaussian functions are calculated using:

$$GF = \left(\sqrt{2\pi} \cdot \frac{FWHM}{2\sqrt{2ln2}}\right)^{-1} \cdot \exp\left(\frac{-(z-z_o)^2}{2(\frac{FWHM}{2\sqrt{2ln2}})^2}\right), \tag{7}$$

where FWHM is the full-width half-maximum, $z$ is the new low-resolution grid point, $z_o$ are the original altitude levels.

Weighting functions were calculated by sampling the GF at the original radiosonde measurement altitude levels and normalizing the GF so that the total weight assigned to all profile elements is equal to one. The weighting functions are different for each pair of coincident profiles because the vertical sampling of each radiosonde profile varies.

Lastly, smoothed profiles were calculated by convolving the water vapour VMR profile of the radiosonde with the weighting functions (wf):

$$x_{smoothed}(z_i) = \sum_{i=1}^{N} wf_i \cdot VMR(z). \tag{8}$$





An example of weighting functions used to align the radiosonde measurement with the approximate vertical resolution of ACE-FTS is shown in Figure 4 (a). Fig. 4 (b) shows an example of a radiosonde profile before and after smoothing. Weighting functions with a *FWHM* equal to 3.0 km have been used to approximate the vertical resolution of ACE-FTS, while comparisons to ACE-MAESTRO, AIRS, and MLS used weighting functions with a *FWHM* of 1.0 km.

## 3.2. Comparison results

Differences between individual coincident profiles were calculated. The means of those differences are presented. When reporting a mean agreement in the text, ± values refer to the standard error in the mean (SEM). Profile results are presented, as well as comparison results at select altitude levels. Results between the satellites and the 125HR at 6.4 km are highlighted because the 125HR has very good sensitivity at that altitude, and this is near the lowermost altitude reached by the ACE measurements. Comparison results between the satellites and the radiosondes are highlighted at 10 km because radiosondes have sensitivity at that altitude and this is the lowermost altitude of other comparison studies, e.g., Sheese et al. (2017), and it is near the lower limit of many satellite datasets.

Some combinations of instruments did not have significant overlap in time, location, or vertical sensitivity. MIPAS and the radiosondes had no coincidences due to a mismatch in the time of day of the measurements as well as the quality control filtering. The temporal ranges of the TES and radiosonde datasets did not overlap. SCIAMACHY did not have any coincidences with the radiosondes, unless the coincidence criterion was expanded to 6 hours. Even then, only 8 matches were found. SCIAMACHY and the 125HR had 201 coincidences; however, SCIAMACHY is limited to altitudes above 10 km, where the 125HR has limited sensitivity.

### 3.2.1. Ground-based reference measurements

As illustrated in Fig. 5, comparison between the 125HR and 137 coincident radiosonde profiles smoothed by 125HR averaging kernels shows agreement within 5% between 8 and 14 km; the 125HR has a wet bias relative to the radiosonde profiles below 8 km of approximately 8% (with closer agreement below 2 km). This is similar to the 6% wet bias in the PEARL 125HR total columns relative to the Eureka radiosondes reported by Weaver et al. (2017). If all possible coincident pairs are used, rather than limiting comparisons to unique pairs, the number of contributing matches increases to 270 and the agreement is very similar.

### 3.2.2. ACE-FTS

76 pairs of coincident ACE-FTS and PEARL 125HR measurements show close agreement. Between 6 and 9 km agreement was within 11 ppmv and 13%; between 8 and 14 km, agreement is within 1.4 ppmv and 10%. Full profile comparisons are shown in Fig. 6. The mean difference of 18 coincident profiles at 6.4 km was − 6.3 ± 8.4 ppmv (0.2 ± 6.8%); the time series of differences at 6.4 km are shown in Fig. 7. At 8.0 km, 46 coincident measurements agreed to within 1.4 ± 2.6 ppmv (7.2 ± 6.6%). Differences at 8.0 km are illustrated in Fig. S1 (a). Correlation plots at 6.4 km, 8.0 km, and 9.8 km are presented in



Fig. 8. Between 6 and 14 km, correlation coefficients ($R$) are between 0.48 and 0.80. Expanding the time criterion to 6 hours nearly doubles the number of coincidences but results in similar agreement. Overall, relative to the 125HR, ACE-FTS shows a wet bias between 8 and 14 km of 7 to 10% and small differences of approximately 10 ppmv (2%) near 6 km (Fig. 6).

108 coincident measurements were found between ACE-FTS and Eureka radiosondes. Profile differences are shown in Fig. 9,
alongside results from other comparisons. These differences are also shown in Fig. S2, where ACE-FTS and ACE-MAESTRO comparison results are presented without other satellites for easier reading. Between 7 and 11 km, differences are within 6 ppmv (12%). At 6 km, ACE-FTS and radiosonde profiles mean differences are $-13.3 \pm 12.1$ ppmv ($22.8 \pm 9.2\%$). Differences at 10 km, $-5.4 \pm 2.0$ ppmv ($-9.1 \pm 6.9\%$), are shown in Fig. 10 (a). Differences at 6 km and 8 km are illustrated in the supplementary materials, Fig. S3 (a) and Fig. S4 (a). Correlation plots at 6.4 km, 8.0 km, and 9.8 km are shown in
Fig. 11. Correlation coefficients between 6 and 12 km range between 0.52 and 0.94.

In addition, comparisons have been done between the ACE-FTS using AIRS as a reference. Differences at 10 km were $-1.5 \pm 0.3$ ppmv ($-6.1 \pm 1.7\%$), increasing at lower altitudes to $-17.0 \pm 3.7$ ppmv ($39.6 \pm 4.3\%$) at 6 km. Correlation coefficients for altitudes between 6 and 12 km were between 0.62 and 0.81. Correlation plots of ACE-FTS vs. AIRS at 6, 8, and 10 km are shown in Fig. 12.

**3.2.3. ACE-MAESTRO**

27 coincident measurements found between ACE-MAESTRO and the PEARL 125HR show agreement within 12 ppmv (7%) between 6 and 8 km and within 3 ppmv (12%) between 9 and 14 km. Overall, between 6 km and 14 km, ACE-MAESTRO shows a dry bias of approximately 10% relative to the 125HR (Fig. 6). Examining the agreement at specific altitudes in the middle and upper troposphere shows scatter around the zero line, illustrated in Fig. 7 and Fig. S1.

20  103 coincident ACE-MAESTRO and radiosonde profiles were found with overlap between 5 and 11 km. Mean differences were large at 5 km, e.g., $-84.0 \pm 121.1$ ppmv ($123.4 \pm 71.1\%$). Percent differences oscillate around $-10\%$ between 7 and 10 km. At 8 km, ACE-MAESTRO had 90 coincidences with the radiosondes, with differences of $-16.3 \pm 8.7$ ppmv and $-7.6 \pm 9.4\%$, shown in Fig. S4. At 10 km, absolute and relative mean differences were $-2.6 \pm 3.2$ ppmv and $-5.9 \pm 10.9\%$, respectively, shown in Fig. 10.

25  In addition, comparisons have been done between the ACE-MAESTRO using AIRS as a reference. Differences at 10 km were $-0.7 \pm 0.9$ ppmv ($-10.5 \pm 3.7\%$), decreasing at lower altitudes to $-13.7 \pm 7.5$ ppmv ($69.9 \pm 13.5\%$) at 6 km. Correlation coefficients for altitudes between 6 and 12 km were about 0.45. Correlation plots of ACE-MAESTRO vs. AIRS at 6, 8, and 10 km are presented in Fig. 12.





### 3.2.4. Other satellite measurements vs. ground-based references

**AIRS**

Close agreement was observed between 3189 coincident AIRS and 125HR measurements and between 2489 coincident AIRS and radiosonde profiles. AIRS profiles agree with the 125HR within 5% between 1 km and 14 km, as shown in Fig. 6. A mean

agreement of $-1.6 \pm 1.5\%$ ($\sigma = 45.9$) was observed between AIRS and 125HR measurements at 6.4 km, where both instruments have good sensitivity. This is shown in Fig. 7 (b). In the mid-troposphere, agreement is within 4%, e.g., at 3.0 km, the mean difference is $-3.8 \pm 1.6\%$ ($\sigma = 32.7$). Correlation coefficients at all altitudes are above 0.84. Correlation plots for AIRS vs. 125HR at 6.4, 8.0, and 9.8 km are shown in Figure 8.

Mean agreement within 5% is observed between AIRS and the radiosondes between 1 and 7 km, as shown in Fig. 9.

Differences as large as 13% are observed between 8 km and 14 km. Differences at 10 km are shown in Fig. 9 (b), where scatter around zero is seen. As well, the time series of differences shows a potential seasonality to the agreement, with a low (dry) bias maximum in summer. Tightening the coincidence criteria to 2 hours and 25 km significantly reduces the number of matches, with 45 contributing to comparisons at 1 km and 1255 contributing to comparisons at 8 km. Results from these tighter matches show differences of less than 4% between 2 and 7 km, with slightly larger differences at 1 km, a mean difference of

$7.5 \pm 0.8\%$ ($\sigma = 30.4$). Differences remained similar between 8 and 14 km with these stricter coincidence criteria.

**MIPAS**

MIPAS v5 and v7 comparisons with the PEARL 125HR show a dry bias of approximately 15% in the upper troposphere. At 6.4 km, the lowest altitude available for comparisons with a reasonable number of coincident measurements ($N = 64$), mean differences using MIPAS v5 were $-22.4 \pm 7.8\%$ ($\sigma = 38.1\%$). MIPAS v7 showed similar differences as v5 with respect to the

125HR at 6.4 km, i.e. $-25.3 \pm 5.9\%$ ($\sigma = 33.5\%$). The time series of differences between the 125HR and MIPAS datasets at 6.4 km is illustrated in Fig. 7 (c), showing large scatter around the zero line. Correlation at 6.4 km was moderate ($R = 0.50$). Between 7 and 14 km a good correlation was observed for both retrieval versions ($R > 0.81$). Agreement improves between 7 and 10 km. MIPAS v5 reaches a mean difference of $-10.1 \pm 1.1\%$ ($\sigma = 27.7\%$) at 9.8 km. Above 10 km, differences are small, better than 2 ppmv and 7%.

No MIPAS measurements were coincident with radiosondes. In part due to the partial overlap of the datasets (September 2008 to April 2012), and also because MIPAS only had Eureka coincidences during mid-day and mid-night, limiting matches within 3 hours of radiosonde launches.

If AIRS is used as a reference, MIPAS v5 and v7 have hundreds or thousands of matches for comparison at each altitude level. The results show that MIPAS has a dry bias relative to AIRS of approximately 15% between 6 and 10 km, comparable to the

125HR results.



**MLS**

Relative to the 125HR, an MLS dry bias is observed in the UTLS, where mean differences range from −18.6 ± 0.8% at 8.8 km to − 0.3 ± 0.4% at 13.6 km. This can be seen in Fig. 6. At 9.8 km, mean differences between 2443 coincidences were − 4.8 ± 0.2 ppmv (−12.5 ± 0.6%); at 12.0 km, mean differences between 2445 coincidences were −0.4 ± 0.0 ppmv (−4.6 ±

5   0.5%).

MLS comparisons with the radiosondes have overlap only between 9 and 13 km; comparisons are shown in Fig. 9. At altitudes between 9 and 12 km the matched measurements are highly correlated, with $R$ values between 0.83 and 0.92. Comparisons between MLS and radiosondes showed a dry bias at altitudes between 8 and 12 km. At 10 km, MLS had 447 coincidences with radiosonde measurements, with a mean differences of −25.6 ± 1.4% (σ = 29.4). The time series of differences between

MLS and the radiosondes at 10 km is shown in Fig. 10 (c).

**SCIAMACHY**

SCIAMACHY could be compared only with the 125HR, as its measurements did not have coincidences with the radiosonde dataset used in this study. 201 SCIAMACHY v3.01 and 1506 SCIAMACHY v4.2 profiles had coincidences with the 125HR; however, these are limited to altitudes above 10 km. Profile comparison results are shown in Fig. 6. For both retrieval versions,

a small (e.g. 5% for v3.01 and 10% for v4.2) dry bias is seen with respect to the 125HR at 10.8 and 12.0 km. At 13.6 km, mean differences were about 1%.

**TES**

TES shows moderate agreement with the PEARL 125HR, but TES had only a single coincidence with the Eureka radiosonde dataset. The latter is largely because TES had no coincidences with Eureka after September 2008, except for a few during mid-

July 2011 (Fig. 1). As shown in Fig. 6, 361 TES measurements showed a dry bias relative to the 125HR of approximately 10% in the lower troposphere, a small dry bias to a small wet bias in the mid-troposphere, and a wet bias (e.g. 20-25%) in the UTLS. The time series of differences at 6.4 km is shown in Fig. 7 (c), where large scatter around the zero line is seen, e.g., σ = 75.1%.

**3.3.   Summary of profile comparisons**

A summary of comparisons between the satellites and the PEARL 125HR is presented in Table 2. Table 3 provides a summary

of the comparisons between the satellites and the Eureka radiosondes. If the distance criterion was reduced to 350 km, similar differences were observed, but with a much smaller number of coincident measurements in some cases. There is no apparent temporal trend in the differences between satellite datasets and the Eureka-based reference measurements.

In addition to the comparison results presented in Fig. 6 through Fig. 12, four figures are presented in the supplementary materials. Fig. S1 shows the time series of differences for the satellite datasets and 125HR at 8 km. Figs. S3 through S5 show

differences between the satellite datasets and the radiosondes at 6, 8, and 12 km altitudes.





In some comparisons, e.g., the comparison between AIRS and the radiosondes at 12 km, the reported mean of the absolute differences and percent differences were different signs, e.g., the mean of the absolute differences was negative while the mean of the percent differences was positive. This is the result of reporting the mean of individual comparisons, rather than comparing the mean profiles of each instrument. The latter would ensure the sign is always the same in both cases. Percent

differences are weighted differently than the absolute differences when the mean is calculated. Histograms were plotted for the differences between each instrument comparison at each altitude discussed in this study. These results (not shown) showed that the differences are typically distributed in a nearly Gaussian manner, justifying the use of the mean, SEM, and standard deviation to characterize the results.

## 4. Discussion

This study's moderately tight temporal criterion, 3 hours, aimed to minimize the impact of water vapour's variability on the observed agreement. The variability of water vapour over the 500 km distance criterion likely contributes to the differences observed between measurements. This is especially true for lower-tropospheric measurements, given the variability of surface terrain in the region around Eureka. The seasonally-changing tropopause height also introduces a source of variability, particularly for altitudes between 8 and 10 km. In the summer, the TPH is often above 8 km at Eureka, and sometimes is above

10 km. $H_2O$ abundances and variability are typically larger at altitudes below the TPH.

Measurement techniques also result in differences in the air sampled. While radiosondes measure air close to Eureka throughout their profile, the 125HR's solar-viewing geometry primarily samples air south of Eureka due to the large SZA of high-latitude measurements. Limb-sounding satellite measurement techniques used by ACE-FTS, ACE-MAESTRO, MIPAS, MLS, and SCIAMACHY yield vertical profiles by observing across long horizontal stretches of atmosphere. While this

technique enables the retrieval to resolve vertical structure, this horizontal path results in profiles containing information about the atmosphere across an extended area. Thus, exact agreement between the satellite and ground-based measurements is not expected. It is worth noting that all of the instruments' measurement techniques observe the atmosphere only in cloud-free conditions, except the Eureka radiosondes.

Since ACE coincidences with Eureka are limited to periods of time when water vapour abundances are relatively similar across

the region, the distance criterion is expected to have less impact on the observed agreement than if year-round measurements were compared. Average March and July water vapour abundances in the area around Eureka are shown in Figure 3. These have been calculated by creating 50 x 50 km grid boxes, centred at Eureka, and taking the mean value of all AIRS profile elements at 400 hPa near Eureka within each box between 2006 and 2016.

Agreement between both ACE instruments and the Eureka reference measurements was closer than that observed in

comparisons conducted by Carleer et al. (2008), which examined an earlier version of these datasets (e.g. ACE-FTS v2.2) and reported differences on the order of 40% at altitudes lower than 15 km and a possible dry bias at around 10 km altitude. Sheese et al. (2017) reported an ACE-FTS negative bias ranging between 3 and 20% relative to MLS and MIPAS at around





14 km; however, the Sheese et al. analysis involves measurements taken over a broad range of global geographic locations and did not discuss altitudes below 13 km.

The ACE-FTS comparisons presented here show a positive (wet) bias of between 7 and 10% relative to the 125HR in the 8 to 14 km altitude range. Relative to the Eureka radiosondes, ACE-FTS shows very close agreement (within 4% or 6 ppmv) in the

upper troposphere (7 to 9 km). At altitudes above 10 km, a positive (wet) bias relative to the radiosondes is observed, ranging between 12 and 32%, although this corresponds to very small mean differences, i.e. of about 1 ppmv. If AIRS is taken as a reference, a larger number of coincidences are found and similar results are observed, although with closer agreement around 10 km. These results indicate ACE-FTS offers accurate $H_2O$ profiles in the Arctic UTLS region, e.g. down to 7 km.

ACE-MAESTRO profiles show a dry bias relative to the 125HR of approximately 10% down to 7 km. Comparisons to the

radiosondes also showed a dry bias, ranging from −3% at 7 km to −21% at 11 km. At 6 km and below, large differences between ACE-MAESTRO and the radiosonde profiles are large, as was the case in the 125HR comparison; however, in both cases there are too few coincidences for firm conclusions. Using AIRS as a reference results in hundreds of coincidences and similar results, e.g. similar magnitudes with an increasingly large difference at altitudes below 7 km.

ACE-MAESTRO shows weak correlations with the Eureka 125HR and radiosonde datasets in Figs. 8 and 11. However, this

is likely due to the combination of water vapour's variability, seen in the Figs. 8 and 11 correlation plots involving AIRS, and the relatively low number of coincidences found. As shown in Fig. 12, the number of coincidences and the correlations between ACE-MAESTRO and AIRS are much larger, e.g. $N = 233$ and $R = 0.64$ at 10 km, while the differences are similar to other comparisons, e.g., there were large differences at 6 km. In addition, the correlation and best-fit line are impacted by outlier points at low altitudes (e.g., at 6.4 km in the comparison with the 125HR) that influence the overall statistics because of the

relatively small number of coincidences at those altitudes. ACE-FTS correlation plots are also affected by outliers.

For both ACE-FTS and ACE-MAESTRO, measurements at altitudes below approximately 5 km are often not possible because ACE's sun-tracker is unable to lock onto the Sun reliably due to cloud effects and refraction (Boone et al., 2005). This issue may contribute to the larger differences observed at low altitudes. This is especially the case with ACE-MAESTRO, whose retrieval produces profiles extending as low as 4 km with tangent heights determined by extrapolation based on the vertical

sampling above 5 km.

AIRS and TES are the only satellite instruments in this study whose measurements are performed in nadir-viewing modes and whose retrieval products reach the lower troposphere. Humidity inversions typically occur near Eureka between 500 m and 2 km in altitude. Sometimes, major structure is seen in the water vapour profile between 2 and 4 km as well. Individual profile-to-profile comparisons with the Eureka radiosondes shows AIRS retrievals do not fully capture structure in the humidity

inversion feature, explaining much of the individual profile differences at the lowest altitude levels. This is expected because the vertical resolution of AIRS is not always sufficient to resolve these vertical structures (Susskind et al., 2014). The AIRS user guide warns of occasional 'strange results' in proximity to near-surface humidity inversions, however, the AIRS profiles





coincident with Eureka showed no features that were oddly shaped or clearly erroneous. The magnitude of the inversion was often inaccurate or the inversion was not seen in the AIRS profile. This could also be in part due to a geographic or temporal mismatch between the measurements.

Similarly, individual profile-to-profile comparisons with the nearest radiosonde profile show TES profiles often capture the
general shape of the lower tropospheric humidity profiles structure; however, the smoothing operation is not enough to bring the measurements into agreement. Where radiosondes from earlier or later in the day reveal a humidity profile with less fine vertical structure, agreement between TES and the 125HR was much closer.

## 5.    Conclusions

This study compared high Arctic UTLS water vapour measurements taken by seven satellite-based instruments with
measurements acquired by the Eureka radiosondes and the PEARL 125HR. The focus of the work was to assess the UTLS water vapour retrieved from ACE-FTS and ACE-MAESTRO measurements. The ACE instruments' ability to observe UTLS water vapour is a valuable contribution to global atmospheric monitoring, as its profiles extend to lower altitudes than many other satellite-based measurements, particularly those retrieved from limb-viewing observations.

ACE-FTS and ACE-MAESTRO showed good agreement with both the radiosondes and the 125HR in the UTLS. No obvious
temporal trend is apparent in the differences. ACE-FTS showed a wet bias of approximately 7 to 10% relative to the 125HR. An ACE-FTS dry bias of 2 to 9% was observed relative to the radiosondes between 8 and 10 km. While agreement is observed in the upper troposphere, the observed agreement did not reach the 5% accuracy goal set by the WMO. ACE-MAESTRO profiles at altitudes below 7 km had large differences relative to both the radiosondes and the 125HR; between 8 and 10  km, a dry bias between 6 and 18% is observed relative to both the radiosondes and the 125HR. Nonetheless, ACE water vapour
measurements showed closer agreement overall with the Eureka reference measurements in the UTLS than did the other satellite datasets examined in this study, with the exception of AIRS.

AIRS water vapour profiles showed close agreement with both the 125HR and radiosonde measurements, i.e. within the 5% WMO target. The observed accuracy of the AIRS measurements suggests they can be used for analysis of humidity conditions near Eureka. Given the high density and frequency of AIRS measurements, it would be worthwhile to use AIRS measurements
to create climatologies of water vapour conditions near the site, and also to examine patterns of water vapour abundances in the region. AIRS data may also be useful for validation studies in cases where radiosonde and 125HR measurements do not offer sufficient numbers of coincident measurements. In addition, global UTLS comparisons between AIRS and ACE water vapour measurements could also be examined to better understand the accuracy of the ACE-FTS and ACE-MAESTRO water vapour datasets.

MIPAS and SCIAMACHY comparisons at altitudes where the data is recommended (i.e., above 10 km) showed agreement within 6% of the 125HR. Coincidences with the radiosondes were not available. At UTLS altitudes where the MIPAS data is not recommended for use, but is included in the publicly available data product, large differences and variability were observed.





This supports the recommendation to limit the use of MIPAS v5 and v7 water vapour profiles to 12 km and above. MIPAS v5 and v7 and SCIAMACHY v3.01 and v4.2 comparison results were very similar.

MLS comparisons with the radiosondes and 125HR between 8 and 12 km showed a dry bias. This aligns with UTLS-region MLS dry biases observed by Hurst et al. (2016) and Vömel et al. (2007b) using FPH measurements.

5   FPH water vapour measurements at Eureka would enhance the ongoing satellite validation work there and enable a valuable reference for PEARL water vapour measurements. FPH measurements would offer the advantage of high accuracy as well as consistent coverage throughout the UTLS. These measurements have been used for the validation of other missions such as MLS (Hurst et al. 2016) and MIPAS (Stiller et al., 2012, using the MOHAVE measurements). Adding FPH measurements would be a useful next step for the comparison and validation of water vapour profiles at Eureka.



Data availability:

Satellite dataset used in this study are available for download through webpages. All require registration except TES and MUSICA.

ACE-FTS and ACE-MAESTRO: http://www.ace.uwaterloo.ca/data.php

AIRS: https://airs.jpl.nasa.gov/data/get_data

MIPAS (IMK retrieval): https://www.imk-asf.kit.edu/english/308.php#org0f1a3a1

MLS: https://mls.jpl.nasa.gov/

SCIAMACHY: http://www.iup.uni-bremen.de/scia-arc/

TES: https://tes.jpl.nasa.gov/data/

The PEARL 125HR water vapour data is available through the online MUSICA repository at:

ftp://ftp.cpc.ncep.noaa.gov/ndacc/MUSICA/.

However, the dataset used in this study has relaxed the usual solar zenith angle criterion to expand available measurements at the high latitude site of Eureka. Please contact the authors regarding access to this dataset.

Radiosonde data used in this study is owned by Environment and Climate Change Canada and is not currently available online.

Please contact the authors regarding access to this dataset.



Author contributions:

D. Weaver led the study, performed 125HR measurements, gathered the datasets, calculated the comparisons between the datasets, created the figures and tables, and wrote the manuscript. K. Strong advised and guided the work and provided significant editing, advice, and comments. K. A. Walker contributed insight into the ACE-FTS dataset, as well as helpful

5    comments on the manuscript. C. Sioris contributed the ACE-MAESTRO water vapour dataset, as well as helpful comments on the manuscript and discussions about the meteorological conditions at Eureka. M. Schneider performed the 125HR retrievals following the MUSICA technique and offered detailed comments. H. Vömel offered insight into the radiosonde data and its use. M. Sommer processed the raw Eureka radiosonde measurement files according to the GRUAN technique. C. T. McElroy contributed insight and comments regarding the ACE-MAESTRO measurement comparisons.

10    J. P. Burrows proposed and leads the SCIAMACHY project, in this context, initiating the concept used for the limb $H_2O$ profiles algorithm, he has contributed to and advised on its evolution, as used in this study. Alexei Rozanov leads the limb retrieval group which has developed key parts of the $H_2O$ retrieval algorithm. Katja Weigel is an expert on limb remote sensing having led the development the V3.1 and V.4 $H_2O$ limb product, validated the data products and initiating and coordinating the IUP contribution to this study.

15    W. G. Read contributed expertise about the MLS dataset. E. Fishbein offered helpful comments on the manuscript and insight regarding the AIRS dataset. G. Stiller offered helpful comments about the manuscript and insight on the MIPAS dataset.

Competing interests:

The authors declare that they have no conflict of interest.





**Acknowledgements.**

This work was primarily funded by the National Sciences and Engineering Research Council (NSERC) through the Probing the Atmosphere of the High Arctic (PAHA) project. Spring visits to PEARL were made as part of the Canadian Arctic ACE/OSIRIS Validation Campaigns funded by the Canadian Space Agency (CSA), with additional support from Environment and Climate Change Canada (ECCC), NSERC, and the Northern Scientific Training Program.

CANDAC/PEARL funding partners are the Arctic Research Infrastructure Fund, Atlantic Innovation Fund/Nova Scotia Research Innovation Trust, Canadian Foundation for Climate and Atmospheric Science, Canada Foundation for Innovation, CSA, ECCC, Government of Canada International Polar Year, NSERC, Ontario Innovation Trust, Ontario Research Fund, Indian and Northern Affairs Canada, and the Polar Continental Shelf Program. This work also received funding from the NSERC CREATE Training Program in Arctic Atmospheric Science, and the CSA-supported Canadian FTIR Observing Network (CAFTON) and Arctic Validation and Training for Atmospheric Research in Space (AVATARS) projects.

The authors would like to thank PEARL PI James Drummond, PEARL Site Manager Pierre Fogal and the CANDAC operators for logistical and operational support at Eureka; ECCC for providing the radiosonde data; Rodica Lindenmaier, Rebecca Batchelor, and Joseph Mendonca for 125HR measurements; and CANDAC Data Manager Yan Tsehtik.

The authors wish to thank the staff at ECCC's Eureka Weather Station for logistical and on-site support.

MUSICA has been funded by the European Research Council under the European Community's Seventh Framework Programme (FP7/2007-2013) / ERC Grant agreement number 256961.

The Atmospheric Chemistry Experiment is a Canadian-led satellite mission mainly supported by the CSA.

We thank the satellite data retrieval and validation teams for the ACE, AIRS, MIPAS, MLS, SCIAMACHY, and TES missions, as well as their funding agencies.

Part of the research was carried out at the Jet Propulsion Laboratory, California Institute of Technology, under a contract with the National Aeronautics and Space Administration.

The SCIAMACHY limb water vapour data sets v3.01 and v4.2 are a result of the DFG (German Research Council) Research Unit "Stratospheric Change and its Role for Climate Prediction" (SHARP) and the ESA SPIN (ESA SPARC Initiative) and SQWG (SCIAMACHY Quality Working Group) projects and were calculated using resources of the German HLRN (High-Performance Computer Center North).

Figures 3, 5, 8, 11, and 12 were produced using python and the matplotlib and numpy packages. The authors would like to thank those development communities.



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



**Table 1: Summary of water vapour datasets used in this study.**

| Type | Satellite/ Location | Instrument | Instrument type | Measurement geometry | Retrieval version | Time range used | Number of measurements within 500 km of Eureka (Aug. 2006 onwards) | Valid altitude range |
|---|---|---|---|---|---|---|---|---|
| Satellite | SCISAT | ACE-FTS | Fourier transform spectrometer | limb | v3.6 | Aug. 2006 - Mar. 2017 | 551 | mid-troposphere to mesosphere |
| | | ACE-MAESTRO | grating spectrometer | limb | v30 | Aug. 2006 - Sept. 2016 | 388 | mid-troposphere to lower stratosphere |
| | AURA | TES | Fourier transform spectrometer | nadir | v6 | Aug. 2006 - July 2011 | 5,630 | P > 100 hPa |
| | | MLS | radiometer | limb | v4.20-v4.22 | Aug. 2006 - Dec. 2015 | 108,072 | P <= 316 hPa |
| | ENVISAT | MIPAS | Fourier transform spectrometer | limb | IMK v5 & v7 | Aug. 2006 - Apr. 2012 | v5: 10428; v7: 10712 | 12 - 50 km recommended; profiles retrieved as low as 4.5 km |
| | | SCIAMACHY | imaging spectrometer | limb | v3.01 & v4.2 | Aug. 2006 - March 2012 | v3.01: 1638; v4.2: 14530 | 11 - 25 km |
| | Aqua | AIRS | grating spectrometer | nadir | v6 | Aug. 2006 - Dec. 2016 | 1,892,348 | P >= 100 hPa |
| Ground-based | PEARL Ridge Lab | 125HR | Fourier transform spectrometer | sun-viewing | MUSICA v2015 | Aug. 2006 - Sept. 2014 (excluding Polar Night) | 1889 (standard); 2713 (no SZA filter) | surface to 14 km (upper altitude varies) |
| | Eureka Weather Station | Radiosondes | capacitance sensor | balloon-borne in situ | Bias corrected and reprocessed | Sept. 2008 - Sept. 2017 | 5515 | surface to 8 - 15 km (upper altitude varies) |



**Table 2: Summary of satellite vs. 125HR comparison results.**

| Instrument (retrieval version) | Altitude [km] | N | mean difference ± SEM [ppmv] | σ [ppmv] | mean difference ± SEM [%] | σ [%] |
|---|---|---|---|---|---|---|
| ACE-FTS (v3.6) | 6.4 | 18 | -6.3 ± 8.4 | 35.7 | +0.2 ± 6.8 | 28.9 |
| | 8.0 | 46 | +1.4 ± 2.6 | 17.6 | +7.2 ± 6.6 | 44.8 |
| | 9.8 | 65 | +0.5 ± 0.4 | 3.3 | +6.1 ± 3.9 | 31.8 |
| | 12.0 | 74 | +0.4 ± 0.1 | 0.9 | +9.7 ± 2.8 | 23.8 |
| ACE-MAESTRO (v30) | 6.4 | 18 | -11.9 ± 16.7 | 71.0 | -6.7 ± 19.2 | 81.5 |
| | 8.0 | 23 | -5.6 ± 6.5 | 31.4 | -6.1 ± 18.7 | 89.6 |
| | 9.8 | 25 | -2.0 ± 1.5 | 7.3 | -10.8 ± 14.7 | 73.6 |
| | 12.0 | 26 | -0.6 ± 0.3 | 1.7 | -11.4 ± 9.5 | 48.5 |
| AIRS (v6) | 3.0 | 434 | -92.8 ± 17.0 | 354.2 | -3.8 ± 1.6 | 32.7 |
| | 6.4 | 881 | -9.7 ± 3.5 | 105.1 | -1.6 ± 1.5 | 75.9 |
| | 8.0 | 1448 | -11.1 ± 1.5 | 56.6 | -2.9 ± 1.0 | 38.8 |
| | 9.8 | 2517 | -2.7 ± 0.2 | 11.4 | -3.5 ± 0.6 | 31.5 |
| | 12.0 | 2798 | -0.1 ± 0.0 | 1.8 | +1.8 ± 0.4 | 23.4 |
| MIPAS (IMK v5) | 6.4 | 24 | -38.2 ± 11.9 | 58.3 | -22.4 ± 7.8 | 38.1 |
| | 8.0 | 93 | -15.8 ± 3.0 | 29.2 | -18.7 ± 3.2 | 30.4 |
| | 9.8 | 604 | -3.6 ± 0.4 | 9.5 | -10.1 ± 1.1 | 27.7 |
| | 12.0 | 897 | -0.3 ± 0.0 | 1.7 | -1.4 ± 0.7 | 21.9 |
| MIPAS (IMK v7) | 6.4 | 32 | -46.9 ± 11.2 | 63.4 | -25.3 ± 5.9 | 33.5 |
| | 8.0 | 96 | -17.0 ± 3.1 | 30.0 | -20.1 ± 3.1 | 29.7 |
| | 9.8 | 634 | -3.8 ± 0.4 | 9.7 | -10.3 ± 1.1 | 27.5 |
| | 12.0 | 902 | -0.3 ± 0.0 | 1.8 | -1.4 ± 0.7 | 22.0 |
| MLS (v4.2) | 8.0 | 13 | -15.9 ± 2.8 | 10.2 | -33.1 ± 3.6 | 12.9 |
| | 9.8 | 2443 | -4.8 ± 0.2 | 11.9 | -12.5 ± 0.6 | 29.7 |
| | 12.0 | 2445 | -0.4 ± 0.0 | 1.9 | -4.6 ± 0.5 | 23.0 |
| SCIAMACHY (IUP v3.01) | 12.0 | 201 | -0.1 ± 0.1 | 1.9 | -1.8 ± 1.9 | 26.7 |
| SCIAMACHY (IUP v4.2) | 12.0 | 1506 | -0.4 ± 0.0 | 1.5 | -5.7 ± 0.5 | 21.2 |
| TES (v6) | 3.0 | 361 | -168.2 ± 45.2 | 859.6 | -1.0 ± 2.3 | 43.4 |
| | 6.4 | 361 | +66.9 ± 15.6 | 296.7 | +23.8 ± 3.9 | 75.1 |
| | 8.0 | 361 | +30.2 ± 5.8 | 110.8 | +27.6 ± 4.0 | 76.2 |
| | 9.8 | 361 | +6.4 ± 1.0 | 19.4 | +26.0 ± 3.2 | 60.4 |
| | 12.0 | 361 | +1.5 ± 0.2 | 3.0 | +23.5 ± 2.1 | 39.2 |



**Table 3: Summary of satellite vs. radiosonde comparison results.**

| Instrument (retrieval version) | Altitude [km] | N | mean difference ± SEM [ppmv] | σ [ppmv] | mean difference ± SEM [%] | σ [%] |
|---|---|---|---|---|---|---|
| ACE-FTS (v3.6) | 6.0 | 57 | -13.3 ± 12.1 | 91.5 | +22.7 ± 9.2 | 69.1 |
|  | 8.0 | 92 | -1.8 ± 3.6 | 34.7 | -1.8 ± 7.2 | 69.5 |
|  | 10.0 | 51 | -5.4 ± 2.0 | 14.0 | -9.1 ± 6.9 | 49.4 |
|  | 12.0 | 19 | +1.2 ± 0.4 | 1.6 | +32.0 ± 6.6 | 28.6 |
| ACE-MAESTRO (v30) | 6.0 | 54 | -62.4 ± 36.8 | 270.7 | +27.0 ± 24.8 | 181.9 |
|  | 8.0 | 90 | -16.3 ± 8.7 | 82.3 | -7.6 ± 9.4 | 89.5 |
|  | 10.0 | 41 | -2.6 ± 3.2 | 20.3 | -5.9 ± 10.9 | 89.5 |
|  | 12.0 | 12 | -1.3 ± 0.6 | 2.0 | -35.8 ± 10.6 | 36.9 |
| AIRS (v6) | 3.0 | 584 | -27.5 ± 16.8 | 407.0 | +5.4 ± 1.9 | 46.2 |
|  | 6.0 | 1423 | -15.6 ± 2.5 | 93.7 | +3.0 ± 1.0 | 39.0 |
|  | 8.0 | 2127 | +3.1 ± 0.9 | 42.3 | +12.7 ± 0.7 | 34.2 |
|  | 10.0 | 868 | -11.2 ± 0.6 | 18.6 | -12.4 ± 0.9 | 27.5 |
|  | 12.0 | 50 | -2.0 ± 1.2 | 8.3 | +5.2 ± 4.1 | 28.8 |
| MLS (v4.2) | 8.0 | 12 | -34.1 ± 28.3 | 98.2 | -25.6 ± 14.8 | 51.1 |
|  | 10.0 | 447 | -5.1 ± 1.2 | 25.0 | -25.6 ± 1.4 | 29.4 |
|  | 12.0 | 42 | -2.4 ± 1.2 | 7.7 | -4.9 ± 4.0 | 26.1 |



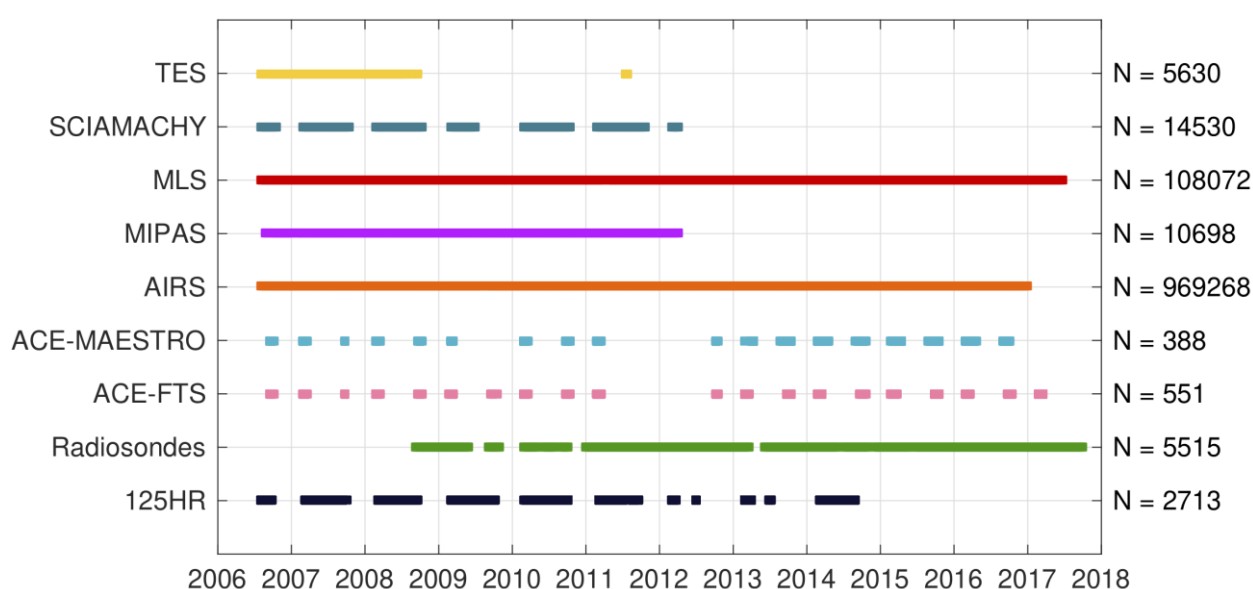

**Figure 1: Temporal range of datasets used in this study.** $N$ is the number of measurements.

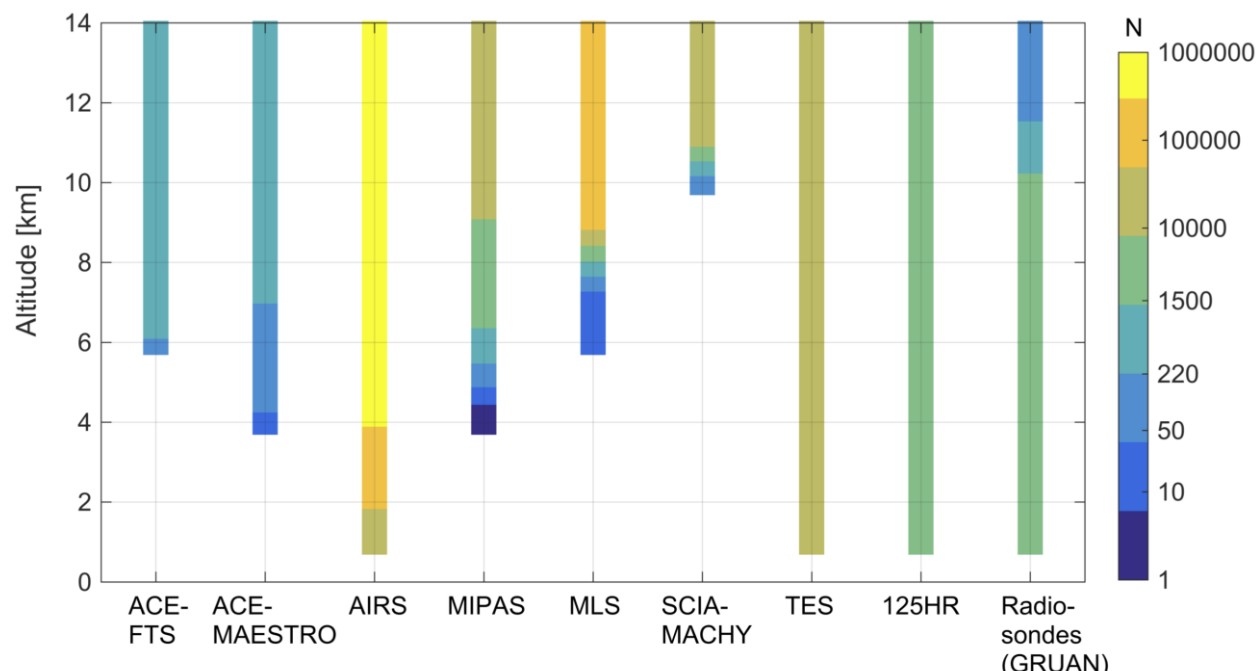

**Figure 2: Vertical range of datasets used in this study. Colour range showing the number of profiles at each altitude level shows the log($N$).**




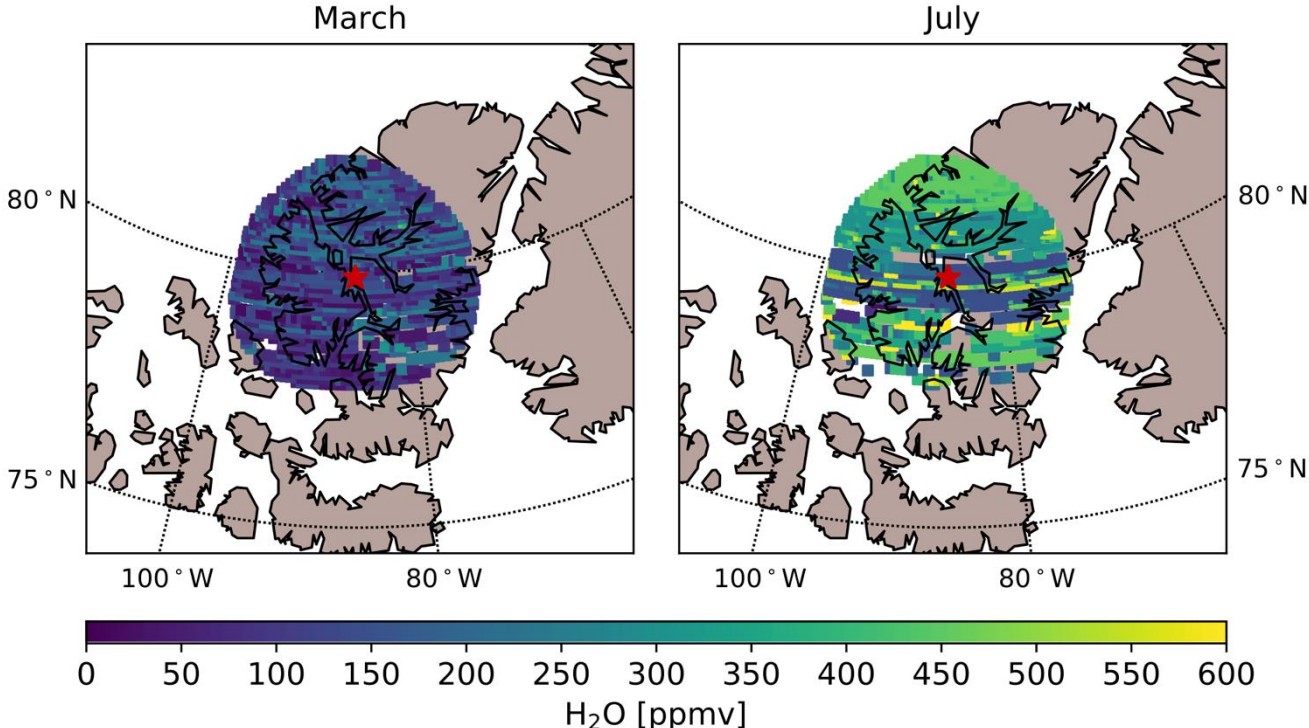

**Figure 3: Average AIRS water vapour abundances at 400 hPa near Eureka (indicated by the red star), calculated by taking the mean value of all AIRS measurements between 2006 and 2016 within 50 x 50 km grid boxes.**





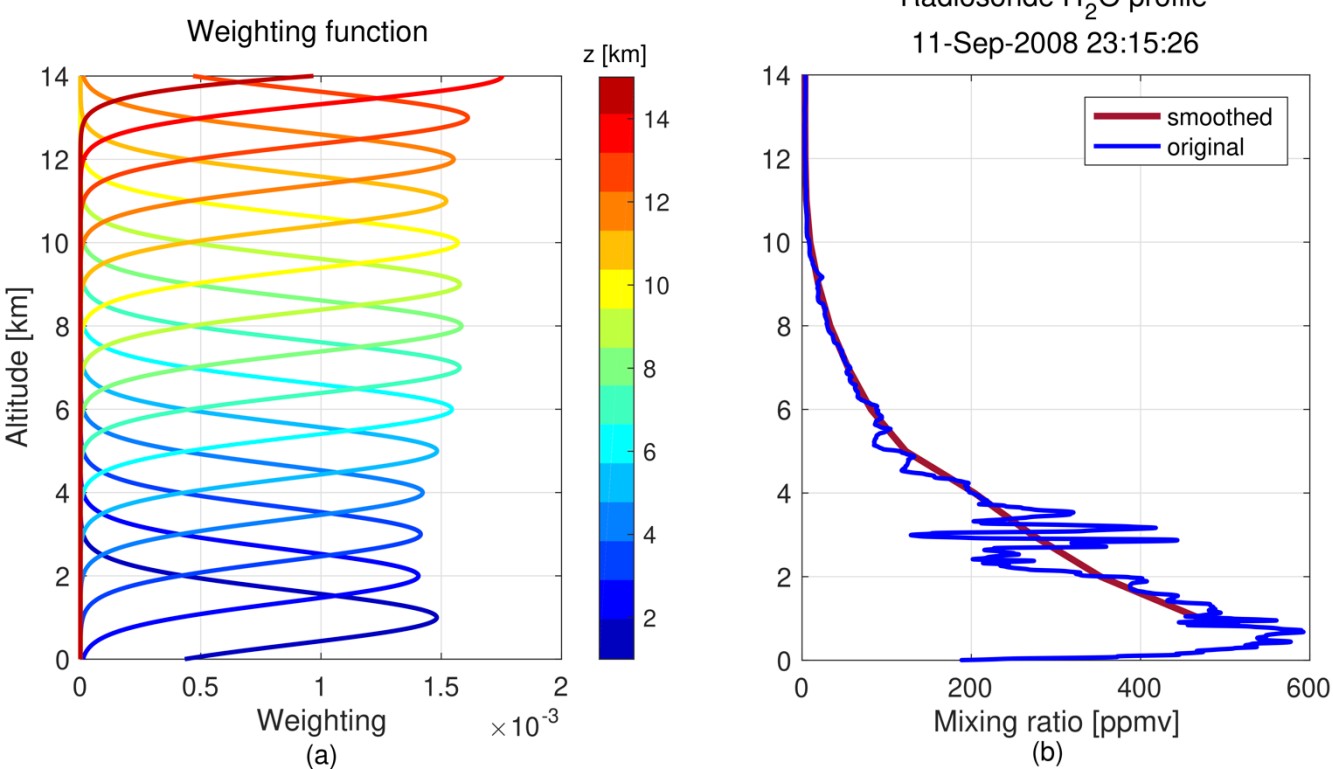

**Figure 4: (a) shows an example of weighting functions used to smooth the radiosonde profiles to ACE-FTS vertical resolution. (b) shows the corresponding radiosonde profile, both as measured (blue line) and after smoothing (maroon line) with the weighting function shown in (a).**





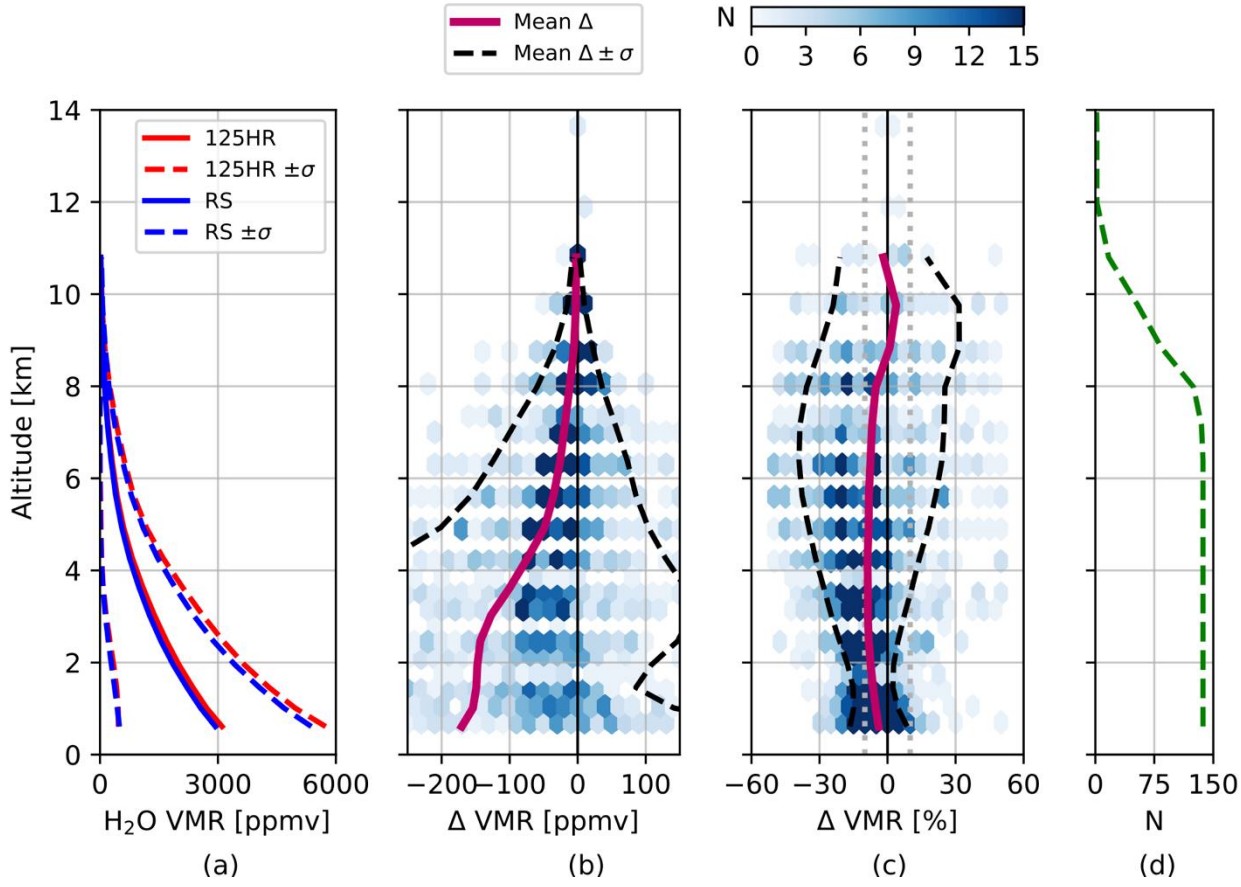

**Figure 5: Comparison between Eureka (GRUAN-processed) radiosonde and PEARL 125HR water vapour VMR. (a) Mean profiles (solid lines) ± the standard deviation (dashed lines). (b) Mean VMR difference (where X = radiosonde and Y = 125HR), using Equation 5.1. (c) Mean percent difference, using Equation 5.2. Grey dotted lines show ±10%. In (b) and (c), the colour shading shows the number (N) of differences in each hexagon. (d) Number of coincident profile pairs at each altitude level.**





**Figure 6: Summary of differences between satellite measurements and PEARL 125HR. (a) The mean of profiles used in the comparison. (b) The mean VMR difference between the satellite profiles and the 125HR profiles, using Equation 5.1. (c) The mean percent difference between the satellite profiles and the 125HR profiles, using Equation 5.2. (d) The number of coincident profile pairs contributing to the comparison at each altitude level. Grey dotted lines in (b) and (c) show ±10 ppmv and ±10%, respectively.**





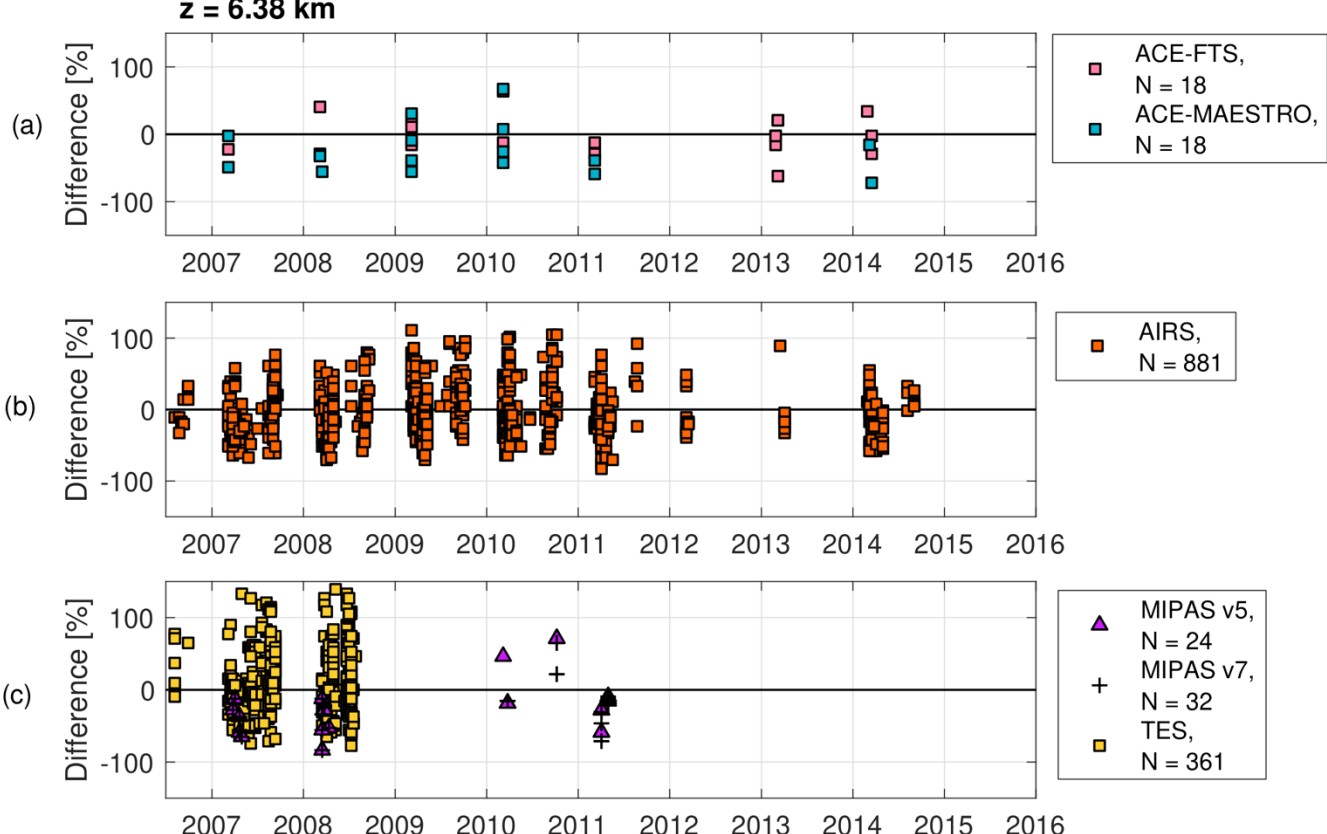

**Figure 7: Time series of percent differences between satellite and 125HR water vapour measurements at 6.4 km altitude for (a) ACE-FTS and ACE-MAESTRO, (b) AIRS, and (c) MIPAS and TES. In each case, the differences follow Eq. 3, where the satellite is *X* and the PEARL 125HR is *Y*.**





**Figure 8: Correlation plots for the ACE-FTS, ACE-MAESTRO, and AIRS satellite measurements vs. 125HR. The number of points in a given hexagon is color-coded to show the density of the points. The scale at each end of a row shows the colour map used for that row. Solid black lines are 1:1 reference lines (i.e. slope = 1); green dashed lines are lines of linear best fit. $N$ is the number of coincident measurements for comparisons between the instruments at that altitude. $R$ is the correlation coefficient. $m$ is the slope of the best fit line.**





**Figure 9: Same as Fig. 6, but a summary of differences between satellite measurements and Eureka radiosondes. A version of this figure with only the ACE-FTS and ACE-MAESTRO is available in the supplementary materials as Fig. S2.**



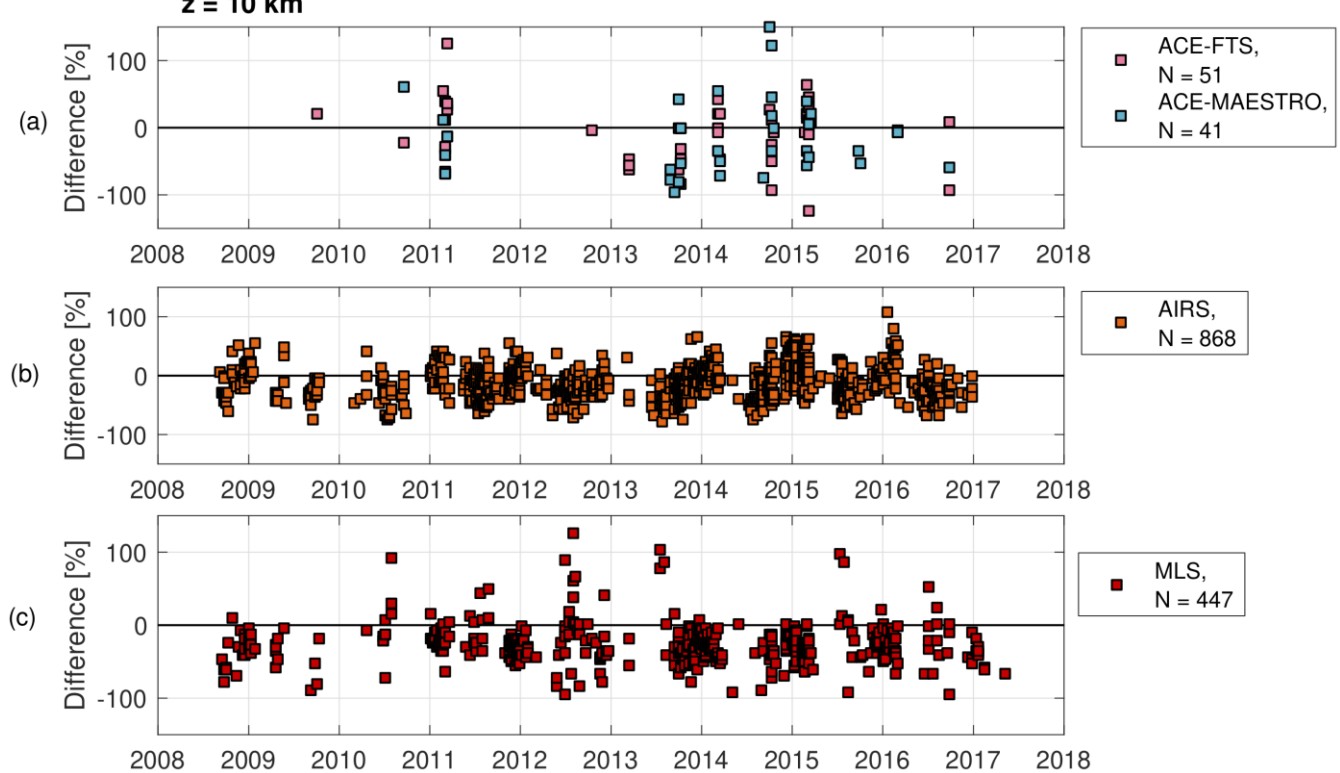

**Figure 10: Time series of percent differences between satellite measurements and the Eureka radiosondes at 10 km altitude for (a) ACE-FTS and ACE-MAESTRO, (b) AIRS, and (c) MLS. In each case, the differences follow Eq. 3, where the satellite is *X* and the Eureka radiosondes is *Y*.**






**Figure 11: Same as Fig. 8, but of correlation plots for the ACE-FTS, ACE-MAESTRO, and AIRS satellite measurements vs. the Eureka radiosondes.**



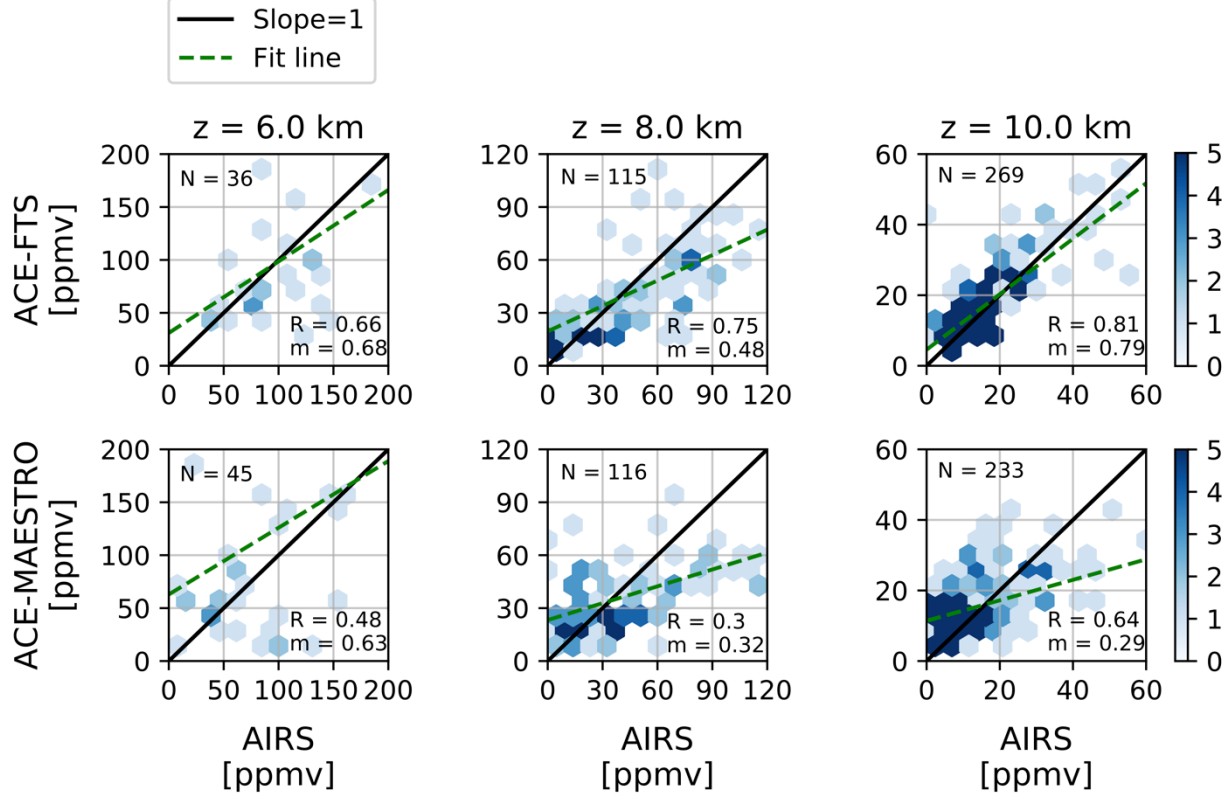

**Figure 12: Same as Fig. 10, but of correlation plots for the ACE-FTS and ACE-MAESTRO vs. the AIRS satellite measurements.**