# Peer review of "Comparison of ground-based and satellite measurements of water vapour vertical profiles over Ellesmere Island, Nunavut"

_Atmospheric Measurement Techniques, 2018_

## Referee Comment (RC1) · Anonymous Referee #1 · 27 Oct 2018

**Overview:**

The primary goal of this research is to compare remotely sensed upper troposphere lower stratospheric water vapour retrievals (UTLS-WV) from the ACE-FTS (ACEF) and ACE-MAESTRO (ACEM) satellite based instruments with that of two ground based water vapour measurement datasets (MIR-FTIR and RH% radiosondes) located at Eureka, Nunavat in the Canadian high arctic. The high artic is a data sparse region and this study provides new additional dataset comparisons. The aim is to see if the ACEF & ACEM datasets are sufficiently accurate (defined as meeting the GCOS 5% accuracy goal for profile measurements), relative to the so called reference ground-based measurements. The secondary aim is also to compare other satellite based UTLS-WV measurements (AIRS, MIPAS, MLS, SCIAMACHY and TES) to the ground based WV datasets and also against ACEF & ACEM. The satellite and MIR-FTIR UTLS-WV datasets are publically available and have been presented in previous published peer reviewed literature and also have been used extensively in other studies. Additionally, in this comparison exercise an extended MIPAS dataset is used, using data from outside the recommended altitude range limits to increase the number of coincidences. Importantly, the RH% radiosonde UTLS WV dataset has been processed using GRUAN procedures. Unfortunately, the current GRUAN dataset is not publically available. There are a total of 9 datasets spanning the time period 2008-2015, with each dataset covering different time periods and altitude ranges. The altitude range comparisons are conducted over are not consistent but with the majority of dataset profile comparisons conducted over the range ~6 to ~12km in ~1km steps. Ubiquitous comparison algorithms are employed based upon spatial and temporal coincidence criteria. Differences relative to the ground based datasets are reported (along with ad-hoc inter-satellite comparisons). Comparisons between the two ground based datasets are also performed. The main findings are that ACEF & ACEM differences relative to the ground based reference measurements are outside the 5% GCOS accuracy goal, but have better comparison statistics compared to other satellites (except for AIRS). The authors conclude AIRS measurements are sufficiently accurate and should be used for future UTLS-WV studies at Eureka (and surrounding region). The authors recommend FPH-WV measurements at Eureka for future ground based-satellite comparisons activities.

The novelty of this study is that is it a comprehensive comparison of UTLS-WV from multiple remote sensing instruments, across multiple platforms in the high artic, and is the first time the Eureka GRUAN processed RH% radiosonde dataset has been used in satellite comparisons. This study will be a welcome addition to literature, especially for the current SPARC WAVAS-II activity. It offers a comparison template which could be extended to other sites with ground-based FTIR or RH% radiosonde measurements. The manuscript meets the scope and requirements of the AMT journal. The manuscript is logically structured. Overall it is well referenced and the writing style is fluent and easily understood.

Improvements can be made. In its current form I do not recommend publication without revision due to a number of issues which are listed below. Such issues need to be addressed: either fixed or with sufficient rebuttal.

**General (major) comments:**

G1/ Possible erroneous values in tables 2 and 3.

As stated in the rapid access review (initial manuscript evaluation) there seems abnormally high amounts of water vapour in the stratosphere (>10ppmv). Quoting this earlier review:

Table 2: MIPAS: 12km: -0.3 ppmv = -1.4% implies a mean VMR of 21.4 ppmv

Table 3: MLS: 12km: -2.4 ppmv = -4.9% implies a mean VMR of 49.0 ppmv

Could the authors please check analysis and table entries and explain the high amounts of water vapour in the lower stratosphere.

G2/ Defining the UTLS and limiting the scope of analysis to the UTLS.

The UTLS altitude range is not defined. Based upon analysis results presented the UTLS has a range of ~6-12 km. There are comparisons made down to ~1km, and up to 14km (fig 5, 6 & 9). Personally, I found that with the multiple datasets and comparisons spanning many altitude ranges it is hard to put together a coherent picture/story. There does seem consistency in comparisons over the 6-12(14) km range, as reflected in tables 2 and 3.

I suggest the scope of the study be limited to the UTLS only, and define the UTLS.  If this approach is taken then the title be changed to reflect the scope. Maybe something like:

"Comparison of ground based-based and satellite measurements of upper troposphere and lower stratosphere water vapour profiles over Ellesmere Island, Nunavut."

G3/ Context

The introduction states the importance and reasons for accurate water vapour measurements in the UTLS. I think there could be more details on the importance of water vapour effects (and changes in water vapour) in the high arctic, hence the importance of the Eureka measurements.

There is a lack of information on past similar multi-measurement campaigns measuring UTLS-WV, such as MOHAVE-2009 (it is mentioned once in the conclusion). Is this current study the first such measurement comparison activity at high latitudes?  I think this would help put this measurement comparison in context.

The first three sentences in the paragraph starting pg 2 line 17 are very weak. They do not add much information. Could such sentences be rewritten with either more information, or a good place to add context as mentioned in the paragraphs above.

G4/ Inclusion of measurement uncertainties.

There is passing mention of measurement uncertainties per instrument (e.g. sondes 3-5%, FTIR ~= 10%) in the text, but this does not carry through in the analysis, figures and tables or in comparison commentary.

For instance in table 2: ACE-FTS: 12km: +0.4 ppmv = 9.7% implies a mean VMR of 4.1 pmv

What are the uncertainties at 12km associated with ACE-FTS and the FTIR measurements? If both were 50% then a 9.7% difference lies within the combined uncertainty. Such uncertainty analysis is not undertaken. Without it, it is hard to put the biases in context of instrument performance. I suggest adding some uncertainty analysis and associated commentary.

Minor, but related points:

-Inclusion of uncertainties estimates (over a given range, per instrument) in table 1 would be helpful.

-In figures 5(c), 6(c) & 9(c) lines are drawn on the +/-10% relative difference. I suspect these have been included as a visual guide. I recommend using lines at 5% (or include lines at 5%) as this is the defined accuracy goal of the study (GCOS goal).

G5/ Layers, vertical resolution, sensitivity and degrees of freedom.

The GRUAN sonde measurements have high vertical resolution with multiple independent data points. For satellite base measurements there is piece-meal mention of vertical resolution (e.g. MIPAS ~3.3km, pg 10, line 17). There is no mention of the FTIR vertical resolution. Linked to vertical resolution, there is only passing mention of the degrees of freedom (DOFs) of the remotely sensed datasets. In the text it quotes TES DOFs to be 3 to 5 (pg 9, line 27), and FTIR retrieval sensitivity is mentioned in section 2.2. I recommend that table 1 be expanded to include columns stating the approximate/average vertical resolution and DOFs for each instrument over the UTLS region. If recommendation S15 (see below) is also implemented (on author discretion) this would also visually indicate vertical resolution to the reader.

Profile comparisons are analysed and reported on ~1km wide altitude layers (table 2 and 3, fig 6 & 9). Given the relatively coarse resolution of the remotely sensed datasets (along with datasets having less degrees of freedom that the number of levels reported on) there will be considerable inter-layer dependence and layer comparison results will be correlated. In figure 5 there seems to be ~28 levels from ~1km to 11km. Given that the Eureka FTIR DOFs are ~1.7 (Schneider, 2016) there is lack of layer independence.

Could authors please comment on inter-layer correlation and performing comparisons using remotely sensed products on vertical grids finer than their associated vertical resolution? Would it be better to perform partial column comparisons (2 or 3 for the UTLS)? This would reduce interlayer correlation.

G6/ Seasonal cycle and seasonality in the TPH

There is no mention of a seasonal dependence in dataset comparisons. All comparisons are made across entire datasets. Looking at Figs 7 and supplementary figures S1 & S3 there seems to be no seasonal bias in comparisons, whilst in Fig 10 (b) there could be a small seasonal bias but nothing mentioned in the manuscript. I think there needs to be a statement or section on seasonal biases (either stating there is a seasonal dependence or not).

There is also no mention on the seasonal variation in the TPH and how this would affect comparisons, especially since the TPH variation could span the current 1km resolution layers. A commentary on TPH height variation in comparisons is required (stating either an impact or lack of impact).

**More specific comments (no particular order):**

S1/ References and referencing:

There is an instance where a paper is referenced in the manuscript, but not in the reference list (Khosrawi, 2018) and conversely there are papers in the reference list, Kurylo, 1991, Sioris, 2016b, &

Stevens, 2013 that are not referenced in the manuscript. Can the authors please recheck the manuscript and reference list to make sure all cross-referencing is correct.

S2/ GCOS and WMO are used interchangeably.

GCOS was referenced in the main part of the manuscript, pg2, line 12, but then subsequent reference to the 5% accuracy goal is attributed to the WMO. Maybe for consistency keep GCOS, not WMO? ...or add a WMO reference.

S3/ Equation 7.

In eqn 7, 'GF' would be better represented as 'GF(z)'.

S4/ Convolving radiosonde VMR profiles with weighting functions: pg 13, line 26 and equation 8.

I think convolving is incorrect terminology, as mathematically it is not a convolution if the weighting function is not static (GF varies with altitude, see fig 4a) and not applicable for instrument averaging kernels. It is also unusual to smooth the high resolution data set (sonde) and report back on the high resolution levels. Usually the smoothed profile is reported on the coarse profile grid. Can the authors please comment on why the smoothed profile is reported back on the high resolution data set levels?

S5/ Equations 2 and 3.

 Minor point: Usually 'X' is the independent variable (ordinate), and 'Y' is the dependent variable (abscissa).  So maybe to hold convention it would be better to have X = reference measurement, Y = satellite measurement (pg 11, line 17). Currently Y = reference measurement.

S6/ Equation 1.

For completeness, the term $e_s(T)$ should be $e_s(T(z))$.

S7/ Sigma ($\sigma$) values in section 3.2.4

There are a series of statistics quoted in section 3.2.4 in which the units are ambiguous, for instance pg 16, line 5:  -1.6 +/-1.5% (sigma = 45.9). What are the sigma units? (I gather ppmv?) Also again on line 7 and line 15.

In line 20, there is a statistic: -25.3 +/- 5.9% (sigma = 33.5%) is '%' the correct unit for the sigma value (the issue also reappears in line 23, and other instances)?

Can I recommend that consistency be preserved in the sense of report statistics in absolute units, i.e. ppmv then as relative (%) in brackets, or vice versa, but not to mix the order up at section level (or even keep consistent across the entire manuscript, if possible)

S8/ Quantifying small dry bias.

Could the 'small dry bias' (pg 17, line 21) be quantified in the text.

S9/ Hexagon symbols in Figs: 5, 8, 11, 12

A pedantic point, sorry, but I'm confused about the use of hexagons as symbols, are these to illustrate a point or an area? I'm assuming data binning, hence its representative of an area. In Cartesian X-Y plotting a hexagon is an interesting choice. Is the data binned within the hexagon region or usual X-Y (rectangular) binning and using the hexagon as a symbol centred in the middle of the rectangular bin?

S10/ Tables 2 and 3

For completeness could SEM be explained in the table captions?

S11/ Figure 3 and accompanying discussion in the text: section 3.1.1

Figure 3 displays a decade of AIRS WV at 400hpa for 2 months: March and July. I'm struggling to find the significance of this figure. On pg 12, line 15 it states that figure 3 shows the spatial variation in water vapour abundance. The data is averaged over 10 years, hence mostly likely averaging out any spatial variation (due to high WV spatial variability). On pg 12, line 18, it states that WV variability is greater in summer (July) than winter (March? or should there be a December or January plot?). Fig 3, 'July', does show larger variability, but stratified in latitudinal bands, is this real? (given the discrete jumps and over 10 years of averaging, I suspect not). There is no commentary on these bands of WV.

At best figure 3 shows a coarse climatology over a large region. Is this what the authors want to convey? If WV seasonal spatial inhomogeneity (i.e. high spatial variance) is to be illustrated then maybe a different visualization should be considered.

S12/ Figure 5.

In figure 5 it seems sonde and FTIR data only goes up to 11km. Is this correct? The text states (pg 5, line 4) that sonde data is limited to less than 15km, but pg 5, line 27 states the mean sonde maximum altitude reached is 11.3km (+/- 4.4km). Also in Fig 6, sonde data is up to 14km. Why is sonde data limited to ~11km in Figure 5? I suspect the number of coincidences above 11km (fig 5, d) is too small.

S13/ Figure 5, part 2…

In the legend it states, X = sonde, Y =125HR. If this refers to use in eqns. 2 and 3 then the FTIR dataset is used as the 'reference' dataset. The sonde dataset would have higher accuracy in the UTLS.  Should the sonde dataset should be used as the reference?

S14/ Table 1. Valid altitude range for SCISAT

The SCISAT valid altitude range table entry is vague, considering ACEF and ACEM are the primary satellite instrument datasets to be investigated. Could a more definite altitude range be specified?

S15/ Displaying instrument vertical resolution.

Figure 4 illustrates ACEF pseudo-vertical resolution (smoothing) and the 'smoothed' radio sonde profile. Figure 4 could be expanded to include the averaging kernels of other instrumentation. This would be helpful in illustrating the comparative vertical resolutions of the different datasets. Looking at figure 2, there seems a brief period in late 2008 that all datasets overlap (or very close to overlapping). A snapshot day of all datasets measurements and vertical resolutions could be displayed as an example. Such a figure could supplant the current figure 4, or be an additional supplementary figure (an idea the authors may wish to consider).

S16/ Two ground based reference datasets

In most studies there is a single defined reference dataset. In this manuscript there are two (FTIR and sonde). I recommend adding a short explanation as to why two reference datasets are used and the consequences of bias between two so called reference datasets (I gather the reason is to get more ground based to satellite coincidences). Given the high vertical resolution and accuracy of the GRUAN sonde dataset (in the UTLS region) should this be the primary (or single) reference dataset?

S17/ Sonde measurements at TPH and above and the recommendation to instigate FPH measurements at Eureka.

In section 2.1 I find a bit of ambiguity. It states that RH% sonde measurements are only valid below the TPH, but then explains that the measurements up to 15km can be used. The sentence on pg 4, line 24 could be changed to state that 'historically' or 'usually' data has been limited to below the TPH, and also referenced as it is an important point.

One of the conclusions of the study is that FPH measurements should be made at Eureka. For UTLS studies, if RH% sonde data is valid up to ~15km then what is gained from FPH measurements, this just needs to be explained a bit more (maybe greater accuracy than the RH% sonde, extended altitude range etc.)? The current sentence on pg 21, line 6 states "FPH measurements would offer the advantage of high accuracy as well as consistent coverage throughout the UTLS". Does this mean sonde data is not consistent? If so, why not?

---

## Referee Comment (RC2) · Anonymous Referee #2 · 1 Nov 2018

Review of "Comparison of ground-based and satellite measurements of water vapour vertical profiles over Ellesmere Island, Nunavut" by D. Weaver et al.

Overview

The goal of this manuscript is to quantify and present the biases between satellite retrievals of water vapor (WV), ground-based WV measurements by FTIR, and balloon borne RH measurements by Vaisala RS92 radiosondes. The study is limited to a single location in the Canadian Arctic where the FTIR and RS92 measurements are performed.

General (Major) Comments

[Figure]

Figures 6 and 9 appear to be identical. It is impossible that they can look exactly the same given what they are meant to show and the obvious differences between the 125HR and RS92 profiles in Figure 5. Also, values stated in the text for specific satellite-RS92 differences don't match up with what's shown in Figure 9. See specific examples below for pages 15 and 16. Finally, Figure 9 shows difference profiles for MIPAS and SCIAMACHY vs RS92 while the text in Section 3.2.4 explicitly says that no MIPAS or SCIAMACHY measurements were coincident with radiosondes. As Figures 6 and 9 are the most important Figures in this paper, it became impossible to continue my review past page 15. My hope is that the authors not only include the correct Figure 9 in the next version, but also take to heart the remainder of my comments and those of the other reviewer(s) that will improve the paper.

I think there are also problems with some of the mean bias values in Tables 2 and 3. For example, for the MIPAS IMK retrievals (v5 and v7) at 12 km in Table 2. The mean difference from the 125HR is given as -0.3 ppmv and -1.4%. If the biases that produce these values are normally distributed, they imply that the mean MIPAS retrieval at 12 km is between 18 and 25 ppmv (-0.25/0.014 and -0.35/0.014). This is way too wet for stratospheric air, and is 3 to 4 times the mean MIPAS IMK retrieval at 12 km (approx. 6 ppmv) shown in Figures 6a and 9a. Another example of this problem is found in Table 3. Mean bias values for AIRS vs RS92 at 12 km are -2.0 ppmv and +5.2%. How can the mean absolute bias (ppmv) and mean relative bias (%) be of opposite signs if the biases are normally distributed? Either there are errors in the mean values presented in these Tables or the distributions of the differences that produce the mean biases are very skewed. If the former, please double check the Table values and make corrections. If the latter, quantifying the biases using Gaussian statistics (i.e., mean and standard error of the mean) is not warranted.

Correlation coefficients and correlation plots are of limited quantitative value in a paper focused on measurement biases between pairs of instruments. Two sets of measurements can be well correlated even though there are huge biases between them! Correlation plots can show biases, but only qualitatively, so consider if the three Figures with correlation plots reveal any quantitative information not already revealed by the profile differences and/or time series of differences. If the correlation plots are deemed unnecessary (my opinion), some (if not all) of the Supplemental Figures could become part of the main manuscript. Please see my specific comments below for Page 15 Line 1 (P15 L1).

The Introduction describes the importance of water vapor in the UTLS and how accurate measurements of WV in the global UTLS are needed. The focus of the paper therefore seems drawn towards WV measurement biases in the UTLS. But this focus becomes lost when you start to compare WV measurements at altitudes as low as 1 km. Why do you apply the same spatial and temporal coincidence criteria to the stratospheric and lower tropospheric data even though the spatiotemporal variability of WV in these regions is very, very different? My advice is to focus this paper on the crucial UTLS region and leave out or downplay the lower tropospheric comparisons.

General Comments (through page 15)

P2 L20 what exactly does "modest vertical resolution" mean? Please be more quantitative here. The vertical resolution of FTIR measurements is very important information for this paper that compares satellite retrievals to the FTIR measurements.

P2 L20-22 Radiosonde humidity sensor measurements also require substantial corrections for solar radiation effects, calibration biases and slow response times in the cold UTLS. It surprises me that frost point hygrometers and lidars are not mentioned here even though the current global coverage of frost point hygrometer sounding sites is starting to surpass the coverage of FTIRs.

P2 L28 "assessing the accuracy and quality" - what does quality mean here if not accuracy?

P3 L1 I believe UT WV measurements will also be compared, not just those in the

stratosphere and lower mesosphere.

P3 L26 move lat/lon to L20 (description of Eureka location)

P4 L23 why is the humidity sensor "no longer able to report a meaningful value"? Is it the cold ambient temperature? Is it the low number density of WV? The solar heating effects on the sensor? Please be more specific.

P5 L2 why describe the Miloshevich et al. (2009) limits when Dirksen et al. (2014) improves the correction algorithms and expands the upper altitude limits of "meaningful" RH measurements by the RS92?

P6 L20 this would be a good place to mention the vertical resolution of the MUSICA FTIR WV profiles

P7 L20 "Correlations between ... were observed to be greater than ..." Why are correlations important in this inter-comparison? Two data sets can be extremely well correlated, even when there is a very large bias between them. Correlation is not a good measure of the agreement between two data sets.

P8 L2 what is the vertical resolution of ACE-MAESTRO WV retrievals in the UT and LS?

P9 L11 what is the vertical resolution of Aura MLS WV retrievals in the UT and LS?

P9 L25 what is the vertical resolution of Aura TES WV retrievals in the UT and LS?

P10 L22 Stiller et al. (2012) compared MIPAS with many types of WV instruments including frost point hygrometers, lidars, microwave radiometers and an FTIR, not just the CFH.

P10 L25 "suggest" and "might be" are very waffly terms. Are there 20-40% biases or not?

P11 L5 Weigel et al. (2016) also compared *SCIAMACHY* v3.01 (not MIPAS v3.01 as

written) to in situ instruments made from balloons (FPH) and aircraft (FISH), not just other satellite retrievals.

P12 L2 Closest in time or space? How did you determine the time stamp for FTIR spectra, which are often co-added for minutes or hours? Also, radiosondes reach 10 km about 30 minutes after they are launched, so how did you set the timestamps for the RS92 profiles?

P12 L9 if the results of comparisons using the closest satellite profile are similar to the results using all coincident profiles, why do you need to show the latter in Supplemental Tables?

P13 L15-18 "... effectively synthesizing a narrow weighting function, then is possible from any one channels. We use of the width ... to estimate a Gaussian smoother generally overestimates ..." These sentences are very poorly constructed. Please fix them.

P14 L4 Above, you stated that the FWHM approximates the vertical resolution of the measurement. So why then do the weighting functions for MLS have a FWHM or 1.0 km when the vertical resolution of MLS retrievals is more like 2-3 km?

P14 L22 Are the 8% and 6% mean differences significantly different from zero? In other words, what are the standard errors of these mean values? It they are not statistically different from zero I would hesitate to call them "biases" because you have no evidence that they are real biases, just mean differences that may equal zero.

P14 L28 I can't see any ACE-FTS differences between 6 and 9 km in Figure 6b that exceed 9 ppmv, so why do you say "was within 11 ppmv"? Also, why report differences for this altitude range when they change from negative to positive at 7 km then become much smaller (in ppmv) and consistent (in ppmv and %) at 8 km and above?

Figure 6 I suggest using fewer red and purple curves, as they are difficult to tell apart. Replace some of them with green, orange and gray. Also, I am guessing that you

[Figure]

discuss satellite-125HR mean differences at 6.4, 8.0 and 9.8 km because these are the altitudes of 125HR retrievals?

P15 L1 and Figure 8 I don't see the value of the correlation coefficients or the correlation plots. The focus of this paper is biases. Correlation coefficients can be near unity when biases between instruments are huge! The correlation plots reveal only qualitative information about biases. For example, the linear fits to ACE-MAESTRO vs 125HR show really awful correlations and essentially no quantitative information about biases. The AIRS panels show good correlations and (qualitatively) that AIRS is biased low at 6.4 and 8.0 km because most of the differences lie below the 1:1 line. What does this Figure (and Figures 11 & 12) show that the vertical profiles of mean differences and time series of differences don't show?

Figure 9 I cannot find a single difference between this Figure and Figure 6, even though they are meant to be showing differences from the RS92 sondes and 125HR, respectively. The two Figures appear to be identical, even when printed, stacked, and held up to backlighting. Are you sure Figures 6 and 9 are actually showing what they are intended to show? The only way they can be exactly the same is if the RS92 and 125HR mean differences are very close to 0 ppmv and 0%, which they are not (Figure 5). The mean differences presented in the text (P15 L7-8) and in Figure 9 do not agree. I suspect Figure 6 appears a second time as Figure 9 in this manuscript.

P15 L19 Your statement here "scatter around the zero line" contradicts what you just concluded, "a dry bias of approx. 10%". The dry bias in ACE-MAESTRO vs 125HR is apparent in Figure 7, so the "scatter" is not "around the zero line" as stated, otherwise there would be no bias.

P16 L10 "Differences as large as 13% are observed between 8 and 14 km." The suspicious Figure 9 shows no relative differences (AIRS-RS92) exceeding 5% between 8 and 14 km.

Unfortunately I cannot continue my review past this point because half of the statements in this very important section are about the biases shown in Figure 9, which is not really Figure 9.

---

## Author Comment (AC1) · 20 Dec 2018

Author response to reviewer's comments on

"Comparison of ground-based and satellite measurements of water vapour vertical profiles over Ellesmere Island, Nunavut"

by Weaver et al.

**Reply to Reviewer #1**

The authors would like to thank reviewer #1 for their attention to detail and helpful comments.

The reviewer's comments are in italics. Replies are in blue.

*G1/ Possible erroneous values in tables 2 and 3.*

*As stated in the rapid access review (initial manuscript evaluation) there seems abnormally high amounts of water vapour in the stratosphere (>10ppmv). Quoting this earlier review:*

*Table 2: MIPAS: 12km: -0.3 ppmv = -1.4% implies a mean VMR of 21.4 ppmv*

*Table 3: MLS: 12km: -2.4 ppmv = -4.9% implies a mean VMR of 49.0 ppmv*

*Could the authors please check analysis and table entries and explain the high amounts of water vapour in the lower stratosphere.*

Water vapour abundances near 20 or 50 ppmv would indeed be well outside expected values in the stratosphere and were not observed in the measurements presented. This can be seen in the panel (a) of the profile comparison figures (i.e., Figures 5, 6, and 9), which show the mean abundances of profiles used for comparisons in this study.

We have calculated the mean absolute difference at each altitude level using:

$$\Delta_{abs}(z) = \frac{1}{N(z)}\sum_{i=1}^{N(z)}[X_i(z) - Y_i(z)], \tag{1}$$

and the mean relative difference using the mean of the percent differences as:

$$\Delta_{rel}(z) = 100\% \times \frac{1}{N(z)}\sum_{i=1}^{N(z)}\frac{[X_i(z)-Y_i(z)]}{Y_i(z)}, \tag{2}$$

rather than calculating the relative difference between the mean profiles using, i.e.:

$$\Delta_{mean}(z) = 100\% \times \frac{\frac{1}{N(z)}\sum_{i=1}^{N(z)}X_i(z) - \frac{1}{N(z)}\sum_{i=1}^{N(z)}Y_i(z)]}{\frac{1}{N(z)}\sum_{i=1}^{N(z)}Y_i(z)} = 100\% \times \frac{\sum_{i=1}^{N(z)}[X_i(z) - Y_i(z)]}{\sum_{i=1}^{N(z)}Y_i(z)}. \tag{3}$$

The absolute difference and percent difference can be combined to calculate the typical abundances only if the percent difference has been derived using the mean profiles of two datasets, e.g. using Equation 3. This cannot be done if the percent difference is derived using the mean of the individual differences and percent differences, e.g., using Equation 2. To ensure the method we used is clear, Equations 1 and 2 have been added to the text of the methods section.

To illustrate the importance of this distinction, let's consider the comparison between MIPAS (IMK v7) and the 125HR at 12 km.

The mean MIPAS abundance was 6.5 ppmv and the mean 125HR abundance was 6.8 ppmv.

Calculating the individual differences between coincident measurements and taking the mean, i.e. applying Equations 1 and 2, results in the following:

$\Delta_{abs}$ (12 km): −0.3 ppmv
$\Delta_{rel}$ (12 km): −1.4%

If these values were combined to calculate 'typical' abundances, the result would be inaccurate and misleading, as pointed out by both reviewers.

If we were instead to apply Equation 3, ie., to calculate the percent differences using the difference between the mean profiles, we get:

$\Delta_{abs}$ (12 km): −0.3 ppmv
$\Delta_{mean}$ (12 km): −4.4%

If we calculate a typical abundance from these values, we get:

$H_2O = \Delta_{abs} / \Delta_{mean} = 0.3$ ppmv $/ 4.4\% = 6.8$ ppmv

This is the original reference value for water vapour abundances, and how the both reviewers expected the numbers to be related.

However, if we examine the mean of the differences, rather than the difference of the means, this calculation of typical abundances is no longer possible.

We could also consider a simple example of two datasets, X and Y, so that the full calculation and numbers can be readily written out:

X = (1, 3, 5)
Y = (2, 2, 8)

The mean of X is: 3
The mean of Y is: 4

The difference between the two means is: −1
The percent difference between the two means ($\Delta_{mean}$) is: −25% (using Y as the reference).

However, we get a different percent difference by taking the mean of the individual percent
differences:

$$\frac{X-Y}{Y} * 100\% = \left(-\frac{1}{2}, \frac{1}{2}, -\frac{3}{8}\right) * 100\% = (-50\%, 50\%, -37.5\%)$$

Mean percent difference ($\Delta_{rel}$) = − 12.5%

Only in the first case, i.e., the percent difference between the means, can the original value be
recovered, i.e.:

−1 / −25% = 4,

i.e., the original mean of Y.

*G2/ Defining the UTLS and limiting the scope of analysis to the UTLS.*

*The UTLS altitude range is not defined. Based upon analysis results presented the UTLS has a*
*range of ~6-12 km. There are comparisons made down to ~1km, and up to 14km (fig 5, 6 & 9).*
*Personally, I found that with the multiple datasets and comparisons spanning many altitude*
*ranges it is hard to put together a coherent picture/story. There does seem consistency in*
*comparisons over the 6-12(14) km range, as reflected in tables 2 and 3.*

*I suggest the scope of the study be limited to the UTLS only, and define the UTLS. If this*
*approach is taken then the title be changed to reflect the scope. Maybe something like:*

*"Comparison of ground based-based and satellite measurements of upper troposphere and*
*lower stratosphere water vapour profiles over Ellesmere Island, Nunavut."*

A definition for the UTLS altitude range has been added, of between 5 and 22 km, in addition to
a definition of the upper troposphere and lowermost stratosphere (UTLMS), i.e., altitudes from 5
km to ~15 km, since the reference instruments have sensitivity only below about this altitude
range.

We prefer not to limit the altitude ranges shown as the tropospheric comparisons add to the
larger story of what measurements are available in this data-poor region. They also put the
UTLMS results in context. As noted in the conclusions, the results usefully motivate further
work with the AIRS dataset.

*G3/ Context*

*The introduction states the importance and reasons for accurate water vapour measurements in the UTLS. I think there could be more details on the importance of water vapour effects (and changes in water vapour) in the high arctic, hence the importance of the Eureka measurements.*

5    *There is a lack of information on past similar multi-measurement campaigns measuring UTLS-WV, such as MOHAVE-2009 (it is mentioned once in the conclusion). Is this current study the first such measurement comparison activity at high latitudes? I think this would help put this measurement comparison in context.*

*The first three sentences in the paragraph starting pg 2 line 17 are very weak. They do not add*
10    *much information. Could such sentences be rewritten with either more information, or a good place to add context as mentioned in the paragraphs above.*

Additional context has been added to motivate the study, including:

"Atmospheric water vapour plays a crucial role in the chemistry, dynamics, and radiative
15    balance of the Earth's atmosphere. Changes to water vapour abundances in the upper troposphere and lower stratosphere (UTLS), which approximately spans altitudes between 5 and 22 km, are particularly consequential for radiative balance (Soden et al., 2008; Riese et al., 2012). Water vapour abundances are expected to increase the most in the lowermost stratosphere (LMS) (Dessler et al, 2013), i.e., altitudes
20    above the tropopause and beneath the tropical tropopause (~17 km), where the radiative impact of additional water vapour is maximum (Solomon et al., 2010). Despite the importance of understanding and monitoring changes to water vapour in this region, accurate long term measurements of water vapour in the upper troposphere and lowermost stratosphere (UTLMS) are limited.
25

….

Satellite-based measurements complement ground-based observations by producing frequent global measurements of atmospheric constituents. More than a dozen satellites
30    are currently (or have been recently) making measurements of water vapour. There is interest in assessing the accuracy and quality of these datasets. The Global Energy and Water Cycle Experiment (GEWEX) (Chahine, 1992) conducted a detailed assessment of tropospheric water vapour measurements. It identified many challenges to attaining a global understanding of the water cycle, including large inconsistencies in long-term total
35    column water vapour measurements in deserts, mountainous regions, and the polar regions (Schröder et al., 2017). The conclusions of the GEWEX review of the state of water cycle measurements reiterated the need to improve on satellite profiling capabilities, diligent validation of data products, and to acquire stable, bias-corrected total column and profile datasets.
40    In addition, a World Climate Research Programme (WCRP) Stratosphere-troposphere Processes And their Role in Climate (SPARC) activity…"

*G4/ Inclusion of measurement uncertainties.*

*There is passing mention of measurement uncertainties per instrument (e.g. sondes 3-5%, FTIR ~= 10%) in the text, but this does not carry through in the analysis, figures and tables or in comparison commentary.*

5    *For instance in table 2: ACE-FTS: 12km: +0.4 ppmv = 9.7% implies a mean VMR of 4.1 pmv*

*What are the uncertainties at 12km associated with ACE-FTS and the FTIR measurements? If both were 50% then a 9.7% difference lies within the combined uncertainty. Such uncertainty analysis is not undertaken. Without it, it is hard to put the biases in context of instrument performance. I suggest adding some uncertainty analysis and associated commentary.*

10    *Minor, but related points:*

*-Inclusion of uncertainties estimates (over a given range, per instrument) in table 1 would be helpful.*

The ACE-FTS dataset does not currently include full uncertainty estimates. The potential for a full uncertainty analysis is limited due to the differences in the information provided by each
15    dataset. For example, ACE-FTS provides an error estimate that represents a statistical fitting error while MLS provides an estimate of the retrieval precision. Other validation work involving ACE datasets, e.g., Sheese et al. (2016), has not used uncertainties to assess the observed biases with other datasets for these reasons. To help inform the bias, the standard error in the mean has been reported, e.g., in Tables 2 and 3.
20    *-In figures 5(c), 6(c) & 9(c) lines are drawn on the +/-10% relative difference. I suspect these have been included as a visual guide. I recommend using lines at 5% (or include lines at 5%) as this is the defined accuracy goal of the study (GCOS goal).*
You are correct: the ±10% relative difference lines were added to aid the reader in interpreting the differences. ±5% lines would helpfully note the GCOS goal; however, when attempted, the
25    scale of the figure made this visually too crowded, particularly Figure 9. Also, the 125HR water vapour profile retrieval's expected accuracy is 10%, making this line meaningful for those comparisons.

*G5/ Layers, vertical resolution, sensitivity and degrees of freedom.*

*The GRUAN sonde measurements have high vertical resolution with multiple independent data points. For satellite base measurements there is piece-meal mention of vertical resolution (e.g. MIPAS ~3.3km, pg 10, line 17). There is no mention of the FTIR vertical resolution. Linked to vertical resolution, there is only passing mention of the degrees of freedom (DOFs) of the remotely sensed datasets. In the text it quotes TES DOFs to be 3 to 5 (pg 9, line 27), and FTIR retrieval sensitivity is mentioned in section 2.2. I recommend that table 1 be expanded to include columns stating the approximate/average vertical resolution and DOFs for each instrument over the UTLS region. If recommendation S15 (see below) is also implemented (on author discretion) this would also visually indicate vertical resolution to the reader.*

*Profile comparisons are analysed and reported on ~1km wide altitude layers (table 2 and 3, fig 6 & 9). Given the relatively coarse resolution of the remotely sensed datasets (along with datasets having less degrees of freedom that the number of levels reported on) there will be considerable inter-layer dependence and layer comparison results will be correlated. In figure 5 there seems to be ~28 levels from ~1km to 11km. Given that the Eureka FTIR DOFs are ~1.7 (Schneider, 2016) there is lack of layer independence.*

*Could authors please comment on inter-layer correlation and performing comparisons using remotely sensed products on vertical grids finer than their associated vertical resolution? Would it be better to perform partial column comparisons (2 or 3 for the UTLS)? This would reduce interlayer correlation.*

The Schneider et al. (2016) paper's 1.7 DOFS refers to a dataset version that is different from the one used in this study. Theirs has a downgraded vertical resolution to align the FTIR $H_2O$ product with the vertical resolution of the retrieved $\delta D$. The $H_2O$ product at Eureka has an average DOFS of 2.9. Barthlott et al. (2017) has a useful table comparing the DOFS of these two versions of the MUSICA water vapour products. The reference at that point in the text has been changed to the Barthlott paper for greater clarity on this point.

Comparisons of partial columns would be much more limited due to the variability of altitude ranges available from many of the datasets, particularly those of primary interest here, i.e., ACE-FTS and ACE-MAESTRO. In addition, the altitude range where radiosonde measurements meet the uncertainty filtering applied in this study varies, often significantly from profile-to-profile, again limiting the ability for partial columns to be compared in the UTLS.

The vertical resolution of the sondes is better than 1 km, e.g. between 10 and 100 m. Each of the satellite datasets are retrieved or measured on a different grid. The comparisons with the radiosondes are reported on a 1-km grid so that a mean difference can be calculated (since this requires a regular grid), and also so that results from different satellite datasets can be compared with the others.

The different vertical resolution of the FTIR and comparison instruments is taken into account by smoothing the satellite profiles with the FTIR averaging kernels prior to the comparison.

*G6/ Seasonal cycle and seasonality in the TPH*

*There is no mention of a seasonal dependence in dataset comparisons. All comparisons are made across entire datasets. Looking at Figs 7 and supplementary figures S1 & S3 there seems to be no seasonal bias in comparisons, whilst in Fig 10 (b) there could be a small seasonal bias but nothing mentioned in the manuscript. I think there needs to be a statement or section on seasonal biases (either stating there is a seasonal dependence or not).*

*There is also no mention on the seasonal variation in the TPH and how this would affect comparisons, especially since the TPH variation could span the current 1km resolution layers. A commentary on TPH height variation in comparisons is required (stating either an impact or lack of impact).*

This is an interesting question. No seasonality was clearly seen in the differences. TPH dependence of the comparisons was plotted but no clear dependence was observed. The first paragraph of the discussion section now notes that "no seasonal pattern in the differences were observed, or pattern with respect to the TPH."

The figure below illustrates an example of the TPH vs. differences figures produced to check for impact on the results:

[Figure]

*Figure 1: ACE-FTS – radiosonde differences at 8 km vs. tropopause height. Points are colour-coded by day of year (DOY). Tropopause height calculated by GRUAN radiosonde processing.*

**Specific comments:**

*S1/ References and referencing:*

*There is an instance where a paper is referenced in the manuscript, but not in the reference list (Khosrawi, 2018) and conversely there are papers in the reference list, Kurylo, 1991, Sioris,*
5  *2016b, & Stevens, 2013 that are not referenced in the manuscript. Can the authors please recheck the manuscript and reference list to make sure all cross-referencing is correct.*

Thank you for catching the referencing mistakes. They have been corrected.

*S2/ GCOS and WMO are used interchangeably.*

*GCOS was referenced in the main part of the manuscript, pg2, line 12, but then subsequent*
10  *reference to the 5% accuracy goal is attributed to the WMO. Maybe for consistency keep GCOS, not WMO? ...or add a WMO reference.*

Agreed. References to WMO has been replaced with GCOS.

*S3/ Equation 7.*
15  *In eqn 7, 'GF' would be better represented as 'GF(z)'.*

Thank you; this change has been made.

*S4/ Convolving radiosonde VMR profiles with weighting functions: pg 13, line 26 and equation 8.*

*I think convolving is incorrect terminology, as mathematically it is not a convolution if the*
20  *weighting function is not static (GF varies with altitude, see fig 4a) and not applicable for instrument averaging kernels. It is also unusual to smooth the high resolution data set (sonde) and report back on the high resolution levels. Usually the smoothed profile is reported on the coarse profile grid. Can the authors please comment on why the smoothed profile is reported back on the high resolution data set levels?*

25  The description of the smoothing procedure in section 3.1 has been modified to state:

"the vertical resolution of radiosonde water vapour VMR profiles were downgraded using the weighting functions"

The radiosonde profiles have variable altitude levels, but measurements are reported roughly every 5 to 10 m in altitude. The satellite datasets all have different, courser, altitude grids. Some
30  of the datasets have different altitude grids from profile-to-profile, e.g., ACE-MAESTRO. A regular grid is needed to put the results on a common basis for comparison. The 1 km grid is a reasonable middle-ground that also allows comparison between the radiosonde and 125HR

results, since many of the 125HR retrieval grid levels of interest are near those values, e.g., 6.4 km, 8.0 km, 9.8 km, and 12.0 km.

*S5/ Equations 2 and 3.*

*Minor point: Usually 'X' is the independent variable (ordinate), and 'Y' is the dependent*
5   *variable (abscissa). So maybe to hold convention it would be better to have X = reference measurement, Y = satellite measurement (pg 11, line 17). Currently Y = reference measurement.*

Satellite – reference is an intuitive way to represent the observed agreement because:

If there is a high bias in the satellite measurement, the difference is positive.
10   If there is a low bias in the satellite measurement, the difference is negative.

This has been used in other validation literature, such as Vömel et al. (2007)'s MLS water vapour validation using cryogenic frostpoint hygrometer measurements.

*S6/ Equation 1.*
15   *For completeness, the term $e_S(T)$ should be $e_S(T(z))$.*

Thank you; this change has been made.

*S7/ Sigma ($\sigma$) values in section 3.2.4*

*There are a series of statistics quoted in section 3.2.4 in which the units are ambiguous, for instance pg 16, line 5: -1.6 +/-1.5% (sigma = 45.9). What are the sigma units? (I gather ppmv?)*
20   *Also again on line 7 and line 15.*

*In line 20, there is a statistic: -25.3 +/- 5.9% (sigma = 33.5%) is '%' the correct unit for the sigma value (the issue also reappears in line 23, and other instances)?*

*Can I recommend that consistency be preserved in the sense of report statistics in absolute units, i.e. ppmv then as relative (%) in brackets, or vice versa, but not to mix the order up at section*
25   *level (or even keep consistent across the entire manuscript, if possible)*

Yes, the units for all standard deviations are the same as the differences preceding them. The text has been updated to ensure that units for the standard deviation are stated explicitly in every instance.

30   The differences reported in section 3 have been updated to ensure that there is consistency in giving absolute units (ppmv) then relative differences (%) in brackets.

*S8/ Quantifying small dry bias.*
*Could the 'small dry bias' (pg 17, line 21) be quantified in the text.*

This has been revised to:

"As shown in Fig. 6, 361 TES measurements showed a dry bias relative to the 125HR of approximately 10% in the lower troposphere, a small dry bias (e.g., −1% at 3.0 km) to a small wet bias in the mid-troposphere (e.g., 3.7% at 3.6 km), and a wet bias (e.g. 20 – 25%) in the UTLS."

*S9/ Hexagon symbols in Figs: 5, 8, 11, 12*

*A pedantic point, sorry, but I'm confused about the use of hexagons as symbols, are these to illustrate a point or an area? I'm assuming data binning, hence its representative of an area. In Cartesian X-Y plotting a hexagon is an interesting choice. Is the data binned within the hexagon region or usual X-Y (rectangular) binning and using the hexagon as a symbol centred in the middle of the rectangular bin?*

The figures using the hexagons (Figs. 5, 8, 11, 12) show the density of the points within the area of the hexagonal symbols. This approach was taken because when plotting points for a correlation figure, the overlap between symbols at each point can mask useful information about how many points are in what location. The plots use hexagons rather than squares because this more closely approximates a circle, allowing the furthest points to be more symmetrically situated with respect to the center (e.g., compared to squares or triangles). The efficiency of this approach, including a comparison and discussion of the use of hexagons vs. other shapes, is described by Carr et al. (1987).

*S10/ Tables 2 and 3*

*For completeness could SEM be explained in the table captions?*

The definition of SEM has been added to the captions for Tables 2 and 3.

*S11/ Figure 3 and accompanying discussion in the text: section 3.1.1*

*Figure 3 displays a decade of AIRS WV at 400hpa for 2 months: March and July. I'm struggling to find the significance of this figure. On pg 12, line 15 it states that figure 3 shows the spatial variation in water vapour abundance. The data is averaged over 10 years, hence mostly likely averaging out any spatial variation (due to high WV spatial variability). On pg 12, line 18, it states that WV variability is greater in summer (July) than winter (March? or should there be a December or January plot?). Fig 3, 'July', does show larger variability, but stratified in latitudinal bands, is this real? (given the discrete jumps and over 10 years of averaging, I suspect not). There is no commentary on these bands of WV.*

*At best figure 3 shows a coarse climatology over a large region. Is this what the authors want to convey? If WV seasonal spatial inhomogeneity (i.e. high spatial variance) is to be illustrated then maybe a different visualization should be considered.*

Figure 3 was indeed included to illustrate the spatial (in)homogeneity of the water vapour abundances in the area around Eureka. March was used because ACE coincidences with Eureka

measurements occurred most often during March. The results for other winter months were not very different. Averaging over the available decade of measurements was intended to provide a general idea of the abundances in the region.

5    This figure has been replaced with a plot showing the Eureka-coincident AIRS measurements at 400 hPa in March and July for a specific representative year (2015) without any binning/averaging. (Plot included below.) There is some overlap between points, but this illustration better conveys the spatial variability of $H_2O$ abundances in the area.

[Figure]

*Figure 2: New "Figure 3" showing the spatial variability of $H_2O$ AIRS measurements at 400 hPa near Eureka for two example months, March and July, in 2015.*

*S12/ Figure 5.*

*In figure 5 it seems sonde and FTIR data only goes up to 11km. Is this correct? The text states*
15    *(pg 5, line 4) that sonde data is limited to less than 15km, but pg 5, line 27 states the mean sonde maximum altitude reached is 11.3km (+/- 4.4km). Also in Fig 6, sonde data is up to 14km. Why is sonde data limited to ~11km in Figure 5? I suspect the number of coincidences above 11km (fig 5, d) is too small.*

Yes, there are no comparisons reported between the FTIR and sonde above 11 km. This is
20    because only altitudes with $N \geq 15$ were shown throughout the study (noted in Section 3.1, which describes the method). Above 11 km, there were only a few coincidences found between those two instruments. This is largely due to the difference in measurement times; the FTIR takes measurements only during daylight (and operator hours emphasize times between 10 AM and 4 PM local time) while the sondes are launched at 6 AM and 6 PM local time (there are
25    occasional exceptions for additional launches). This is illustrated in the figure below, a histogram

that shows the available MUSICA measurement times by the hour of the day. Daily radiosonde launch times are noted with red dashed lines. Atypical occasional radiosonde launch times are noted with blue dashed lines.

[Figure]

Figure 3: Histogram of Eureka MUSICA measurement times. Red dashed lines indicate typical daily radiosonde launch times (6 AM and 6 PM). Blue dashed lines indicate occasional atypical radiosonde launch times (12 AM and 12 PM).

In addition, the sonde measurements are filtered by uncertainty, which removes many of the measurements above 10 km. Text noting that mean profiles are not plotted for z > 11 km because N < 15 at those altitudes has been added to the Figure 5 caption.

*S13/ Figure 5, part 2...*

*In the legend it states, X = sonde, Y =125HR. If this refers to use in eqns. 2 and 3 then the FTIR dataset is used as the 'reference' dataset. The sonde dataset would have higher accuracy in the UTLS. Should the sonde dataset should be used as the reference?*

That is true. However, the difference in results would be only the sign of the statistics. This arrangement was chosen for consistency with other comparisons to the 125HR, as the radiosondes in this case are smoothed using the 125HR averaging kernels.

*S14/ Table 1. Valid altitude range for SCISAT*

*The SCISAT valid altitude range table entry is vague, considering ACEF and ACEM are the primary satellite instrument datasets to be investigated. Could a more definite altitude range be specified?*

5    The altitude range reported in Table 1 is worded in this manner because the valid altitude range of the ACE instruments is varies greatly from measurement-to-measurement (e.g., some ACE-FTS profiles extend only to 15 km at their lowest; in other cases, they extend to 5.5 km). In addition, determining the lowest altitude range where measurements are accurate is one of the objectives of the study.

10   *S15/ Displaying instrument vertical resolution.*

*Figure 4 illustrates ACEF pseudo-vertical resolution (smoothing) and the 'smoothed' radio sonde profile. Figure 4 could be expanded to include the averaging kernels of other instrumentation. This would be helpful in illustrating the comparative vertical resolutions of the different datasets. Looking at figure 2, there seems a brief period in late 2008 that all datasets*
15   *overlap (or very close to overlapping). A snapshot day of all datasets measurements and vertical resolutions could be displayed as an example. Such a figure could supplant the current figure 4, or be an additional supplementary figure (an idea the authors may wish to consider).*

This is an interesting idea. An examination of all dataset coincidences resulted in one specific day where all datasets had measurements coincident with the 125HR and a few days where all
20   datasets had measurements coincident with the radiosondes. The former and an example of the latter are shown below in Figures 6 and 7. They have been added as supplemental figures.

With regard to the vertical resolutions of the datasets, there will be an examination and presentation of this in a forthcoming WAVAS-II paper by Walker and Stiller. We aimed to avoid
25   overlap with their efforts.

[Figure]

*Figure 4: Individual satellite vs. 125HR profile comparisons on March 12, 2008.*

[Figure]

*Figure 5: Individual satellite vs. radiosonde comparisons on March 09, 2014.*

*S16/ Two ground based reference datasets*

*In most studies there is a single defined reference dataset. In this manuscript there are two (FTIR and sonde). I recommend adding a short explanation as to why two reference datasets are used and the consequences of bias between two so called reference datasets (I gather the reason is to get more ground based to satellite coincidences). Given the high vertical resolution and accuracy of the GRUAN sonde dataset (in the UTLS region) should this be the primary (or single) reference dataset?*

It is true that most studies use a single reference dataset. Two datasets were used in this study for a few reasons:

- Two datasets are available. The sondes and the FTIR are the only instruments routinely producing water vapour profiles from a standardized methodology at Eureka at the moment.
- The best available reference, the GRUAN-processed radiosondes, does not have ideal overlap with all the satellite measurements. In large part, this is due to the time of day they are launched and their twice-per-day frequency of measurements. In addition, the available raw data files needed for GRUAN processing have gaps and are available only from mid-2008 onwards. Consequently, some comparisons with the radiosondes are limited in time, space, or altitude-ranges. Too few coincidences were found between the radiosondes and MIPAS, SCIAMACHY, and TES for meaningful comparisons, for example.
- The GRUAN processing is not part of an ongoing arrangement, since Eureka is not an official GRUAN site. It is useful to see how well the FTIR comparison results align with the GRUAN results for ongoing monitoring of water vapour profiles produced by satellite instruments that have coincidences with Eureka.

Text has been added to the start of section 3 commenting on the use of two reference datasets.

> "Water vapour profiles from ACE-FTS, ACE-MAESTRO, AIRS, MIPAS, MLS, SCIAMACHY, and TES were compared with Eureka radiosonde and PEARL 125HR measurements following the methodology described below. Two ground-based reference measurements are used in this study to maximize comparisons with available satellite measurements. The radiosondes provide high vertical resolution profiles; however, they had few or no coincidences with MIPAS, SCIAMACHY, and TES. The 125HR, while having more limited vertical resolution, had coincident measurements with all satellite datasets used in this study."

*S17/ Sonde measurements at TPH and above and the recommendation to instigate FPH measurements at Eureka.*

*In section 2.1 I find a bit of ambiguity. It states that RH% sonde measurements are only valid below the TPH, but then explains that the measurements up to 15km can be used. The sentence on pg 4, line 24 could be changed to state that 'historically' or 'usually' data has been limited to below the TPH, and also referenced as it is an important point.*

The suggested change, inserting 'usually', has been made.

*One of the conclusions of the study is that FPH measurements should be made at Eureka. For UTLS studies, if RH% sonde data is valid up to ~15km then what is gained from FPH measurements, this just needs to be explained a bit more (maybe greater accuracy than the RH% sonde, extended altitude range etc.)? The current sentence on pg 21, line 6 states "FPH measurements would offer the advantage of high accuracy as well as consistent coverage throughout the UTLS". Does this mean sonde data is not consistent? If so, why not?*

Where we say that the radiosondes do not offer consistent coverage, that refers to the availability of the radiosonde profiles in the UTLS. Profiles are only sometimes used in that region due to the uncertainty filtering applied. The greater accuracy and lower uncertainty of the FPH measurements would be an advantage, as would their ability to capture information at higher altitudes in the lower stratosphere. The wording has been changed to more clearly articulate that it is the altitude range, in addition to the better accuracy, that would be an advantage of the FPH:

> "FPH water vapour measurements at Eureka would enhance the ongoing satellite validation work there and enable a valuable reference for PEARL water vapour measurements. FPH measurements would offer improved accuracy as well better coverage throughout UTLS altitudes relative to the radiosondes and 125HR. FPH measurements have been used for the validation of other missions such as MLS (Hurst et al. 2016) and MIPAS (Stiller et al., 2012, using the MOHAVE measurements). Adding FPH measurements would be a useful next step for the comparison and validation of water vapour profiles at Eureka."

**References**

Carr, D. B., Littlefield, R. J., Nicholson, W. L., and Littlefield, J. S.: Scatterplot Matrix Techniques for large N, Journal of the American Statistical Association, 82 (389), pp. 424-436, doi:10.1080/01621459.1987.10478445, 1987.

Vömel, H., Barnes, J. E., Forno, R. N., Fujiwara, M., Hasebe, F., Iwasaki, S., Kivi, R., Komal, N., Kyrö, E., Leblanc, T., B. Morel, B., Ogino, S.-Y., Read, W. G., Ryan, S. C., Saraspriya, S., Selkirk, H., Shiotani, M., J. Valverde Canossa, and D. N. Whiteman: Validation of Aura Microwave Limb Sounder water vapor by balloon-borne Cryogenic Frost point Hygrometer measurements, *J. Geophys. Res.*, 112, D24S37, doi:10.1029/2007JD008698, 2007.

---

## Author Comment (AC2) · 20 Dec 2018

Author response to reviewer's comments on

"Comparison of ground-based and satellite measurements of water vapour vertical profiles over Ellesmere Island, Nunavut"

by Weaver et al.

**Reply to Reviewer #2**

The authors would like to thank reviewer #2 for their attention to detail and helpful comments.

The reviewer's comments are included in italics. Replies are in blue.

Major comments:

*Figures 6 and 9 appear to be identical. It is impossible that they can look exactly the same given what they are meant to show and the obvious differences between the 125HR and RS92 profiles in Figure 5. Also, values stated in the text for specific satellite-RS92 differences don't match up with what's shown in Figure 9. See specific examples below for pages 15 and 16. Finally, Figure 9 shows difference profiles for MIPAS and SCIAMACHY vs RS92 while the text in Section 3.2.4 explicitly says that no MIPAS or SCIAMACHY measurements were coincident with radiosondes. As Figures 6 and 9 are the most important Figures in this paper, it became impossible to continue my review past page 15. My hope is that the authors not only include the correct Figure 9 in the next version, but also take to heart the remainder of my comments and those of the other reviewer(s) that will improve the paper.*

**(1)** The correct version of Figure 9 was included in the initial submission of the manuscript during submission; however, minor modifications to improve the readability were suggested during the technical review. When updating the file for re-submission, the lead author mistakenly included a second copy of Figure 6 where Figure 9 should have been. This has been corrected, and should satisfy the other concerns raised about consistency between the text and figure. We apologize for this unfortunate mistake.

*I think there are also problems with some of the mean bias values in Tables 2 and 3. For example, for the MIPAS IMK retrievals (v5 and v7) at 12 km in Table 2. The mean difference from the 125HR is given as -0.3 ppmv and -1.4%. If the biases that produce these values are normally distributed, they imply that the mean MIPAS retrieval at 12 km is between 18 and 25 ppmv (-0.25/0.014 and -0.35/0.014). This is way too wet for stratospheric air, and is 3 to 4 times the mean MIPAS IMK retrieval at 12 km (approx. 6 ppmv) shown in Figures 6a and 9a. Another example of this problem is found in Table 3.*

Water vapour abundances near 20 or 50 ppmv would indeed be well outside expected values in the stratosphere and were not observed in the measurements presented. This can be seen in the

panel (a) of the profile comparison figures (i.e., Figures 5, 6, and 9), which show the mean abundances of profiles used for comparisons in this study.

We have calculated the mean absolute difference at each altitude level using:

$$\Delta_{abs}(z) = \frac{1}{N(z)} \sum_{i=1}^{N(z)} [X_i(z) - Y_i(z)], \tag{1}$$

and the mean relative difference using the mean of the percent differences as:

10    $$\Delta_{rel}(z) = 100\% \times \frac{1}{N(z)} \sum_{i=1}^{N(z)} \frac{[X_i(z) - Y_i(z)]}{Y_i(z)}, \tag{2}$$

rather than calculating the relative difference between the mean profiles using, i.e.:

$$\Delta_{mean}(z) = 100\% \times \frac{\frac{1}{N(z)} \sum_{i=1}^{N(z)} X_i(z) - \frac{1}{N(z)} \sum_{i=1}^{N(z)} Y_i(z)]}{\frac{1}{N(z)} \sum_{i=1}^{N(z)} Y_i(z)} = 100\% \times \frac{\sum_{i=1}^{N(z)} [X_i(z) - Y_i(z)]}{\sum_{i=1}^{N(z)} Y_i(z)}. \tag{3}$$

15

The absolute difference and percent difference can be combined to calculate the typical abundances only if the percent difference has been derived using the mean profiles of two datasets, e.g. using Equation 3. This cannot be done if the percent difference is derived using the
20    mean of the individual differences and percent differences, e.g., using Equation 2. To ensure the method we used is clear, Equations 1 and 2 have been added to the text of the methods section.

To illustrate the importance of this distinction, let's consider the comparison between MIPAS (IMK v7) and the 125HR at 12 km.
25
The mean MIPAS abundance was 6.5 ppmv and the mean 125HR abundance was 6.8 ppmv.

Calculating the individual differences between coincident measurements and taking the mean, i.e. applying Equations 1 and 2, results in the following:
30
        $\Delta_{abs}$ (12 km): −0.3 ppmv
        $\Delta_{rel}$ (12 km): −1.4%

If these values were combined to calculate 'typical' abundances, the result would be inaccurate
35    and misleading, as pointed out by both reviewers.

If we were instead to apply Equation 3, ie., to calculate the percent differences using the difference between the mean profiles, we get:

40        $\Delta_{abs}$ (12 km): −0.3 ppmv
        $\Delta_{mean}$ (12 km): −4.4%

If we calculate a typical abundance from these values, we get:

$H_2O = \Delta_{abs} / \Delta_{mean} = 0.3$ ppmv $/ 4.4\% = 6.8$ ppmv

This is the original reference value for water vapour abundances, and how the both reviewers expected the numbers to be related.

However, if we examine the mean of the differences, rather than the difference of the means, this calculation of typical abundances is no longer possible.

We could also consider a simple example of two datasets, $X$ and $Y$, so that the full calculation and numbers can be readily written out:

$X = (1, 3, 5)$
$Y = (2, 2, 8)$

The mean of $X$ is: 3
The mean of $Y$ is: 4

The difference between the two means is: $-1$
The percent difference between the two means ($\Delta_{mean}$) is: $-25\%$ (using $Y$ as the reference).

However, we get a different percent difference by taking the mean of the individual percent differences:

$$\frac{X - Y}{Y} * 100\% = \left(-\frac{1}{2}, \frac{1}{2}, -\frac{3}{8}\right) * 100\% = (-50\%, 50\%, -37.5\%)$$

Mean percent difference ($\Delta_{rel}$) $= -12.5\%$

Only in the first case, i.e., the percent difference between the means, can the original value be recovered, i.e.:

$-1 / -25\% = 4,$

i.e., the original mean of $Y$.

*.... Mean bias values for AIRS vs RS92 at 12 km are -2.0 ppmv and +5.2%. How can the mean absolute bias (ppmv) and mean relative bias (%) be of opposite signs if the biases are normally distributed? Either there are errors in the mean values presented in these Tables or the distributions of the differences that produce the mean biases are very skewed. If the former,*
5 *please double check the Table values and make corrections. If the latter, quantifying the biases using Gaussian statistics (i.e., mean and standard error of the mean) is not warranted.*

The distributions of the differences are generally Gaussian. For example, Figure 1 shows a
10 histogram of the differences between AIRS and 125HR measurements at 6.4 km.

[Figure]

15 *Figure 1: Histogram of differences between AIRS and 125HR water vapour measurements at 6.4 km. The dashed red line is the mean of the differences; the blue dashed lines show one standard deviation above and below the mean. The solid tan line shows the median of the differences.*

20

25

In a few cases, the sign of the absolute and percent differences are not the same. There are a few reasons for this.

In some cases, e.g., AIRS vs. radiosondes comparison at 12 km, the number of coincidences is relatively small (N = 50). As the number of coincidences becomes small, we expect the approximation of a Gaussian distribution to be less justified. Indeed, in the case of ACE-FTS vs. 125HR at 6.4 km, the standard error in the mean indicates the mean absolute and percent differences, which are in this case of opposite sign, are not significantly distinct from zero.

In other cases where there is large number of coincidences and a roughly Gaussian distribution, there is a small skewness in the distribution that differs enough between the absolute differences and percent differences that the means land on opposite sides of zero. The skewness is not large, but has this effect because the mean of the overall distribution is close to zero relative to the range of values involved. Histograms of absolute differences and percent differences between AIRS and GRUAN at 6.4 km are shown in Figure 2 and 3. These illustrate how the small differences in the skewness of the absolute difference and percent difference distributions, nearly centered at zero, can have means with opposite signs. Also note that the medians of the absolute differences (Figure 2) and percent differences (Figure 3) have the same sign (negative).

If the median differences and percent differences are examined, all comparisons have the same sign at all altitudes. Median differences have been added to Table 2 and 3, which summarize the results.

[Figure]

*Figure 2: Histogram of absolute differences between AIRS and GRUAN-processed radiosonde water vapour measurements at 6 km. Lines defined as in Figure 1.*

[Figure]

*Figure 3: Histogram of percent differences between AIRS and GRUAN-processed radiosonde water vapour measurements at 6 km. Percent differences calculated using: (AIRS – GRUAN)/GRUAN * 100%. Lines defined as in Figure 1.*

*Correlation coefficients and correlation plots are of limited quantitative value in a paper focused on measurement biases between pairs of instruments. Two sets of measure- ments can be well correlated even though there are huge biases between them! Cor relation plots can show biases, but only qualitatively, so consider if the three Figures with correlation plots reveal any quantitative information not already revealed by the profile differences and/or time series of differences. If the correlation plots are deemed unnecessary (my opinion), some (if not all) of the Supplemental Figures could become part of the main manuscript. Please see my specific comments below for Page 15 Line 1 (P15 L1).*

It is true that correlation coefficients need to be carefully interpreted. In this study, they are used in combination with the differences to show how well the measurements agree. In particular, the correlation plots illustrate how closely the measurements agree and how much variation in the differences exist, i.e., the spread in the values. This information is also shown in the difference timeseries; however, it is useful to examine the datasets as a whole – e.g., not as a timeseries. This can reveal, for example, if there are measurement biases or differences that affect measurements at larger vs. smaller abundances. The use of these plots is common in the validation literature, e.g., with the FTIR MUSICA product (Schneider et al., 2010), other water vapour measurement techniques (Buehler et al., 2012), ACE and OSIRIS satellite products (Adams et al., 2012), and other satellite missions such as GOSAT (Frankenberg et al., 2013; Ohyama et al., 2017) and MLS (Vömel et al., 2007). The use of the correlation coefficient as a

part of an overall assessment of agreement between datasets has been even more widely used, e.g., for comparisons between ACE-FTS profiles and other satellite datasets (Sheese et al., 2016).

5 *The Introduction describes the importance of water vapor in the UTLS and how accurate measurements of WV in the global UTLS are needed. The focus of the paper therefore seems drawn towards WV measurement biases in the UTLS. But this focus becomes lost when you start to compare WV measurements at altitudes as low as 1 km. Why do you apply the same spatial and temporal coincidence criteria to the stratospheric and lower tropospheric data even though*
10 *the spatiotemporal variability of WV in these regions is very, very different? My advice is to focus this paper on the crucial UTLS region and leave out or downplay the lower tropospheric comparisons.*

One of the key questions to be answered in the work is "how low, in altitude, can ACE profile
15 measurements of water vapour be trusted?" This question necessarily involves measurements as low as 4.5 km (in the case of ACE-MAESTRO). While the paper could exclude profiles with values below this altitude, it is useful to see the comparisons at all available altitudes for the retrieved profiles of the datasets used so that the context of the observed agreement at altitudes of particular interest are interpreted in their full context. For example, if the AIRS measurements
20 were to suddenly diverge from the radiosondes at 4 km, and show a large bias in the lower troposphere, their agreement in the upper troposphere would be placed in a different context than the consistent agreement observed throughout the troposphere in this study. Also, including these available results gives a more complete assessment of what vertical profiles are available at Eureka. Moreover, the results at tropospheric altitudes motivate a study that focuses on the use of
25 AIRS water vapour data in the high Arctic.

That said, it is certainly true that the spatio-temporal variability increases greatly at lower altitudes with important consequences for the selection of coincidence criteria. For that reason (and others), tighter coincidence criteria were examined. In Section 3.2.4, which discusses the
30 AIRS comparison results, the paper notes that a much tighter coincidence criteria of 25 km and 2 hours shows similar comparison results.

General Comments:
35
*P2 L20 what exactly does "modest vertical resolution" mean? Please be more quantitative here. The vertical resolution of FTIR measurements is very important information for this paper that compares satellite retrievals to the FTIR measurements.*

40 The vertical resolution of the FTIR measurements varies; the mean DOFS are 2.9 for the PEARL 125HR MUSICA product. This has been added to the text in Section 2.2's description of the dataset used in this study, as suggested in the comment for P6 L20.

*P2 L20-22 Radiosonde humidity sensor measurements also require substantial corrections for solar radiation effects, calibration biases and slow response times in the cold UTLS. It surprises me that frost point hygrometers and lidars are not mentioned here even though the current global coverage of frost point hygrometer sounding sites is starting to surpass the coverage of*

5 *FTIRs.*

The paper's introduction focused on the approaches used in this study and those that are most widely used. While frostpoint hygrometers (FPHs) offer definite advantages over radiosondes and FTIR spectrometers, their geographic deployment is much less widespread than radiosondes.

10 In addition, they typically acquire measurements less often than radiosondes and FTIRs, i.e., some sites launch them only monthly, and their data timeseries are usually shorter. Moreover, there have been no FPH measurements taken from Eureka. This is regrettable. The nearest sites where FPH measurements are taken are Ny Ålesund, Svalbard, and Barrow, Alaska, which are both roughly 2000 km away. For these reasons, FPH measurements are noted in the conclusions

15 as a promising area of future work, as it would be valuable to add them to the suite of instruments at PEARL/Eureka.

*P2 L28 "assessing the accuracy and quality" - what does quality mean here if not accuracy?*

20 "Quality" has been removed as redundant, as suggested.

*P3 L1 I believe UT WV measurements will also be compared, not just those in the stratosphere and lower mesosphere.*

25 This sentence has been reworded to include the upper troposphere.

*P3 L26 move lat/lon to L20 (description of Eureka location)*

This has been done.

30

*P4 L23 why is the humidity sensor "no longer able to report a meaningful value"? Is it the cold ambient temperature? Is it the low number density of WV? The solar heating effects on the sensor? Please be more specific.*

35 Original text in that paragraph and the following paragraph notes that Miloshevich (2009) shows that the RS92 radiosonde capacitance sensor responds accurately at low temperatures (–70°C) and at low abundances (5 ppmv) but that low pressures are a limiting factor.

*P5 L2 why describe the Miloshevich et al. (2009) limits when Dirksen et al. (2014) improves the correction algorithms and expands the upper altitude limits of "meaningful" RH measurements by the RS92?*

Dirksen et al. (2014) improves the correction algorithms, but the resulting GRUAN data product does not set out upper altitude limits. This motivates the use of a filtering approach for this study based on the calculated uncertainties resulting from the Dirksen analysis technique. In a few cases, the uncertainty of the GRUAN-processed humidity profile remains below the filtering threshold well above 15 km, e.g., to 25 km. Out of an abundance of caution, the altitude limit suggested by Miloshevich et al., 100 hPa, on the radiosonde measurements was cited and applied as an additional quality control filter, resulting in any profile at Eureka that passes the uncertainty filtering being limited to a maximum height of 15 km, which is approximately the altitude of 100 hPa. This is also roughly the boundary for the upper troposphere and lowermost stratosphere, the area of specific interest of this study.

*P6 L20 this would be a good place to mention the vertical resolution of the MUSICA FTIR WV profiles*

This has been added:

"The mean degrees of freedom for signal (DOFS) of the Eureka MUSICA retrievals are 2.9."

*P7 L20 "Correlations between ... were observed to be greater than ..." Why are correlations important in this inter-comparison? Two data sets can be extremely well correlated, even when there is a very large bias between them. Correlation is not a good measure of the agreement between two data sets.*

Correlations have been discussed above in the reply to the major comments.

*P8 L2 what is the vertical resolution of ACE-MAESTRO WV retrievals in the UT and LS?*
*P9 L11 what is the vertical resolution of Aura MLS WV retrievals in the UT and LS?*
*P9 L25 what is the vertical resolution of Aura TES WV retrievals in the UT and LS?*

Approximate values are given in the instrument descriptions as available.

The ACE-MAESTRO vertical resolution is approximately 1 km.

The MLS 4.2.x product document states that the vertical resolution of the water vapour profiles is 1.3 – 3.6 km between 316 and 0.22 hPa. The altitudes used in this study are 316 hPa and the levels immediately above it, putting the resolution closest to the 1.3 km end of the range.

TES vertical resolution varies by altitude, latitude and species. The DOFS have been improved in the most recent version (6) used here, with DOFS between 3 and 5. However, at polar latitudes, in the UTLS, the vertical resolution is 11.6 km, while in the troposphere it is 6.0 km (Worden et al., 2004).

The text has been revised as follows:

> "The ACE-MAESTRO water vapour retrieval algorithm produces profiles with an approximate vertical resolution of 1 km, and is described by Sioris et al. (2010) with updates described in Sioris et al. (2016)."
>
> ….
>
> "MLS water vapour profiles are vertically resolved at pressures less than 383 hPa, with a vertical resolution ranging between 1.3 and 3.6 km from 316 to 0.22 hPa (Livesey et al., 2016)."
>
> ….
>
> "The vertical information content of TES profiles varies; retrievals with less than 3 DOFS are filtered out. In the subset of measurements examined in this study, TES DOFS range between 3.0 and 5.2. At polar latitudes, the vertical resolution is approximately 11.6 km between 400 and 100 hPa and 6.0 km between 1000 and 400 hPa (Worden et al., 2004)."

*P10 L22 Stiller et al. (2012) compared MIPAS with many types of WV instruments including frost point hygrometers, lidars, microwave radiometers and an FTIR, not just the CFH.*

While the Stiller et al. (2012) study included comparisons to other instruments, the comparison to the CFH was most relevant to the discussion here, as it was the best reference measurement.

*P10 L25 "suggest" and "might be" are very waffly terms. Are there 20-40% biases or not?*

Conclusive statements regarding the bias of an instrument cannot be derived from comparisons at a single site. The term 'suggest' is intended to convey that these specific results are to be interpreted in the context of the wider validation literature. The specific use of these terms in this instance reflect the terms used by Stiller et al. to describe the results of the cited work.

*P11 L5 Weigel et al. (2016) also compared \*SCIAMACHY\* v3.01 (not MIPAS v3.01 as written) to in situ instruments made from balloons (FPH) and aircraft (FISH), not just other satellite retrievals.*

Thanks - correction made.

*P12 L2 Closest in time or space? How did you determine the time stamp for FTIR spectra, which are often co-added for minutes or hours? Also, radiosondes reach 10 km about 30 minutes after they are launched, so how did you set the timestamps for the RS92 profiles?*

The closest pair in time were kept. The timestamp for the FTIR spectra were the scan start time. Scans took about 5 minutes, following standard NDACC procedures and settings. The timestamp for radiosondes was the launch time. These clarifications have been added to the respective descriptions of the datasets.

*P12 L9 if the results of comparisons using the closest satellite profile are similar to the results using all coincident profiles, why do you need to show the latter in Supplemental Tables?*

The comparisons using all coincident profiles was offered in the supplemental materials for reader's interest, to demonstrate the accuracy of the statement that the results are similar (they are not identical), and to provide a complete record that might be useful for future studies that might want to compare results that use this approach rather than the paired approach used in the main manuscript.

*P13 L15-18 "... effectively synthesizing a narrow weighting function, then is possible from any one channels. We use of the width ... to estimate a Gaussian smoother generally overestimates ..." These sentences are very poorly constructed. Please fix them.*

The first sentence has been removed while the second has been revised to be:

"We use of the width of the AIRS weighting functions to estimate a Gaussian smoothing width that generally overestimates the amount of smoothing."

*P14 L4 Above, you stated that the FWHM approximates the vertical resolution of the measurement. So why then do the weighting functions for MLS have a FWHM or 1.0 km when the vertical resolution of MLS retrievals is more like 2-3 km?*

The MLS data quality document specifies (page 66) that the vertical resolution of the water vapour profile ranges from 1.3 – 3.6 km from 316 – 0.22 hPa. The altitudes of interest are at the highest pressure (lowest altitude) of that range, thus closer to 1 km than the 3-4 km typical of stratospheric altitudes.

*P14 L22 Are the 8% and 6% mean differences significantly different from zero? In other words, what are the standard errors of these mean values? It they are not statistically different from zero I would hesitate to call them "biases" because you have no evidence that they are real biases, just mean differences that may equal zero.*

The standard errors on the approximate 8% difference between the 125HR and the radiosondes under 8 km altitude ranged between 1 and 3%, suggesting a real difference. SEM values are provided both in the text when specific altitude results are given, and also in the summary of results in Tables 2 and 3. In addition, inspection of individual coincident profiles frequently show a negative RS – 125HR difference. However, caution in this result is justified, given that the expected accuracy of this FTIR water vapour profile retrieval is approximately 10%. Additional text has been added to clarify the standard errors and remind the reader of the expected precision of the FTIR profiles.

*P14 L28 I can't see any ACE-FTS differences between 6 and 9 km in Figure 6b that exceed 9 ppmv, so why do you say "was within 11 ppmv"? Also, why report differences for this altitude range when they change from negative to positive at 7 km then become much smaller (in ppmv) and consistent (in ppmv and %) at 8 km and above?*

−11.0 ppmv is the difference between ACE-FTS and 125HR at 5.6 km altitude. The text has been revised to state they agree within the suggested 9 ppmv in the 6 − 9 km altitude range. This range had been reported for comparison with other instruments. The text states that the differences are smaller above 8 km, i.e., "between 8 and 14 km, agreement is within 1.4 ppmv and 10%".

*Figure 6 I suggest using fewer red and purple curves, as they are difficult to tell apart. Replace some of them with green, orange and gray. Also, I am guessing that you discuss satellite-125HR mean differences at 6.4, 8.0 and 9.8 km because these are the altitudes of 125HR retrievals?*

In this study, each instrument is given a colour, which is used consistently across all figures. The suggested colours are used for other instruments, some of which are not in this figure, but are in others. For consistency across figures, the colours have been kept as they are.

Yes, 6.4, 8.0, and 9.8 km are altitudes from the FTIR retrieval grid. This has been noted in Section 3.1.2, in the description of the method:

> "Comparisons between satellite measurements and the FTIR are thus presented on the MUSICA retrieval altitude grid, e.g., 6.4 km, 8.0 km, and 9.8 km."

*P15 L1 and Figure 8 I don't see the value of the correlation coefficients or the correlation plots. The focus of this paper is biases. Correlation coefficients can be near unity when biases between instruments are huge! The correlation plots reveal only qualitative information about biases. For example, the linear fits to ACE-MAESTRO vs 125HR show really awful correlations and essentially no quantitative information about biases. The AIRS panels show good correlations and (qualitatively) that AIRS is biased low at 6.4 and 8.0 km because most of the differences lie below the 1:1 line. What does this Figure (and Figures 11 & 12) show that the vertical profiles of mean differences and time series of differences don't show?*

Correlations have been discussed above in the reply to the major comments.

*Figure 9 I cannot find a single difference between this Figure and Figure 6, even though they are meant to be showing differences from the RS92 sondes and 125HR, respec- tively. The two Figures appear to be identical, even when printed, stacked, and held up to backlighting. Are you sure Figures 6 and 9 are actually showing what they are intended to show? The only way they can be exactly the same is if the RS92 and 125HR mean differences are very close to 0 ppmv and 0%, which they are not (Figure 5). The mean differences presented in the text (P15 L7-8) and in Figure 9 do not agree. I suspect Figure 6 appears a second time as Figure 9 in this manuscript.*

This correction has been made and was discussed above in the reply to the major comments.

*P15 L19 Your statement here "scatter around the zero line" contradicts what you just concluded, "a dry bias of approx. 10%". The dry bias in ACE-MAESTRO vs 125HR is apparent in Figure 7, so the "scatter" is not "around the zero line" as stated, otherwise there would be no bias.*

This sentence has been revised.

*P16 L10 "Differences as large as 13% are observed between 8 and 14 km." The suspicious Figure 9 shows no relative differences (AIRS-RS92) exceeding 5% between 8 and 14 km.*

This disconnect is due to the aforementioned Figure 9 issue, which has been corrected.

**References**

Adams, C., Strong, K., Batchelor, R. L., Bernath, P. F., Brohede, S., Boone, C., Degenstein, D., Daffer, W. H., Drummond, J. R., Fogal, P. F., Farahani, E., Fayt, C., Fraser, A., Goutail, F., Hendrick, F., Kolonjari, F., Lindenmaier, R., Manney, G., McElroy, C. T., McLinden, C. A., Mendonca, J., Park, J.-H., Pavlovic, B., Pzamino, A., Roth, C., Savastiouk, V., Walker, K. A., Weaver, D., Zhao, X.: Validation of ACE and OSIRIS ozone and $NO_2$ measurements using ground-based instruments at 80°N. Atmospheric Measurement Techniques, *5*(5), 927–953. https://doi.org/10.5194/amt-5-927-2012, 2012.

Buehler, S. A., Östman, S., Melsheimer, C., Holl, G., Eliasson, S., John, V. O., Blumenstock, T., Hase, F., Elgered, G., Raffalski, U., Nasuno, T., Satoh, M., Milz, M., & Mendrok, J.: A multi-instrument comparison of integrated water vapour measurements at a high latitude site. Atmospheric Chemistry and Physics, *12*(22), 10925–10943. https://doi.org/10.5194/acp-12-10925-2012, 2012.

Frankenberg, C., Wunch, D., Toon, G., Risi, C., Scheepmaker, R., Lee, J. E., Wennberg, P., & Worden, J.: Water vapor isotopologue retrievals from high-resolution GOSAT shortwave infrared spectra. Atmospheric Measurement Techniques, *6*(2), 263–274. https://doi.org/10.5194/amt-6-263-2013, 2013.

Ohyama, H., Kawakami, S., Shiomi, K., Morino, I., & Uchino, O.: Intercomparison of $XH_2O$ data from the GOSAT TANSO-FTS (TIR and SWIR) and ground-based FTS measurements: Impact of the spatial variability of $XH_2O$ on the intercomparison. *Remote Sensing*, *9*(1). https://doi.org/10.3390/rs9010064, 2017.

Schneider, M., Romero, P. M., Hase, F., Blumenstock, T., Cuevas, E., & Ramos, R.: Continuous quality assessment of atmospheric water vapour measurement techniques: FTIR, Cimel, MFRSR, GPS, and Vaisala RS92. *Atmospheric Measurement Techniques*, *3*(2), 323–338. https://doi.org/10.5194/amt-3-323-2010, 2010.

Sheese, P. E., Walker, K. A., Boone, C. D., McLinden, C. A., Bernath, P. F., Bourassa, A. E., Burrows, J. P., Degenstein, D. A., Funke, B., Fussen, D., Manney, G. L., Thomas McElroy, C., Murtagh, D., Randall, C. E., Raspollini, P., Rozanov, A., Russell, J. M., Suzuki, M., Shiotani, M., Urban, J., von Clarmann, T., and Zawodny, J. M.: Validation of ACE-FTS version 3.5 NO*y*
5    species profiles using correlative satellite measurements. *Atmospheric Measurement Techniques*, *9*(12), 5781–5810. https://doi.org/10.5194/amt-9-5781-2016, 2016.

Shephard, M. W., Herman, R. L., Fisher, B. M., Cady-Pereira, K. E., Clough, S. A., Payne, V. H., Whiteman, D. N., Comer, J. P., Vömel, H., Miloshevich, L. M., Forno, R., Adam, M., Osterman, G. B., Eldering, A., Worden, J. R., Brown, L. R., Worden, H. M., Kulawik, S. S.,
10   Rider, D. M., Goldman, A., Beer, R., Bowman, K. W., Rodgers, C. D., Luo, M., Rinsland, C. P., Lampel, M., and Gunson, M. R.: Comparison of Tropospheric emission spectrometer nadir water vapor retrievals with in situ measurements. Journal of Geophysical Research Atmospheres, *113*(15), 1–17, https://doi.org/10.1029/2007JD008822, 2008.

15   Vömel, H., Barnes, J. E., Forno, R. N., Fujiwara, M., Hasebe, F., Iwasaki, S., Kivi, R., Komal, N., Kyrö, E., Leblanc, T., B. Morel, B., Ogino, S.-Y., Read, W. G., Ryan, S. C., Saraspriya, S., Selkirk, H., Shiotani, M., J. Valverde Canossa, and D. N. Whiteman: Validation of Aura Microwave Limb Sounder water vapor by balloon-borne Cryogenic Frost point Hygrometer measurements, J. Geophys. Res., 112, D24S37, doi:10.1029/2007JD008698, 2007.

20   Weaver, D., Strong, K., Schneider, M., Rowe, P. M., Sioris, C. E., Walker, K. A., Mariani, Z., Uttal, T., Thomas McElroy, C., Vömel, H., Spassiani, A., & Drummond, J. R.: Intercomparison of atmospheric water vapour measurements at a Canadian High Arctic site. Atmospheric Measurement Techniques, *10*(8), 2851–2880. https://doi.org/10.5194/amt-10- 2851-2017, 2017.

---

## Author Response (AR2)

Author response to reviewer's comments on:

"Comparison of ground-based and satellite measurements of water vapour vertical profiles over Ellesmere Island, Nunavut" by Weaver et al. (2018)

*Reviewer comments are in italics; replies are in blue text.*

**Reply to Reviewer #3**

The authors would like to thank reviewer #3 for their attention to detail.

*P9, L28: Pressure is usually denoted by a small letter, thus it should read here p < 100 hPa*
*P10, L29: timeframe -> time frame (?)*
*References. Please check these. I may be that there are more errors than the three I found:*
*P26, L13: Atmo-spheric -> Atmospheric*
*P27, L11: Hopfner -> Höpfner*
*P28, L16: H"opfner -> Höpfner*

Thank you, these corrections have been made.

**Reply to Reviewer #4**

The authors would like to thank reviewer #4 for their attention to detail and helpful comments.

*Review of "Comparison of ground-based and satellite measurements of water vapour vertical profiles over Ellesmere Island, Nunavut" by Weaver et al.*

*The manuscript presents an assessment of ACE-FTS and ACE-MAESTRO water-vapour products in the upper troposphere - lower stratosphere with two different reference datasets: in situ measurements from radiosondes and ground-based sounding with upward looking FTIR.*
*In addition, the study involves other satellite humidity products, from AIRS, MIPAS, MLS, SCHIAMACHY and TES.*

*The assessment is limited to the Arctic station at Eureka and aims at evaluating the suitability of such products for climate purposes, considering GCOS requirements. The study holds on about 7-years worth of data.*

*The study is found of great relevance for the scientific community, in characterising the performances and limitations of those products in view of climate applications, process studies...*

*The manuscript is found very well structured and written, with clear description of the various datasets (satellite and reference) and with good presentation material for the results. QA practices regarding the reference datasets and data selection is sufficiently described or referenced. The methodology for comparing the profiles involves averaging kernels as much as*

*possible, or proxy-kernels where these are not available.*

*I recommend the publication of the manuscript, with few small comments.*

*General:*

*The study makes use of data from one location only, and hence - despite the quality of the exhaustive work - there are limitations to the generalisation of the conclusions wrt objectives: i.e. assess the relevance of the proposed ACE datasets in view of climate applications with GCOS requirements. The authors are invited to expand a little bit on this in their conclusions and provide outlook to possible future study/assessment plans seeking larger coverage in the Northern Polar cap if that is the main region of interest.*

Text has been added to the conclusion section, specifically identifying the opportunity for more comprehensive monitoring and analysis of water vapour in the Arctic using the satellite datasets discussed in this study – AIRS in particular.

> "Future work with these satellite datasets could involve an analysis of water vapour abundances in the UTLS across the Arctic, e.g., using ACE measurements. Moreover, the density of measurements and close agreement between AIRS and the Eureka GRUAN-processed radiosonde dataset motivates the use of the AIRS dataset to investigate water vapour abundances across the Arctic throughout the troposphere."

*Since the point of the study, besides documenting the performances of the various satellite products, is to evaluate the performances wrt GCOS requirements, it is important to quantify the contribution of the collocation uncertainties to the overall budget (satellite - sonde/125HR). The issue is briefly touched on in the manuscript but should be discussed a bit more in my view: the conclusion of the study seems to be that ACE products are not suitable according to GCOS requirements. They might be more relevant than it appears in fact if the collocation uncertainties were considered.*

The impact of collocation distances was mentioned in a few places in the manuscript. For example, it is noted in Section 3.2.4 that the AIRS comparisons were conducted at a variety of close coincidence criteria, e.g., 2 hours and 25 km. The results of these close coincidences were similar to the results reported in the main discussion, figures, and data tables. However, it was not possible to analyze the impact of the coincidence criteria with other instruments, e.g., ACE, because there were not enough matches at close coincidence values. For example, fewer than 15 ACE-FTS and GRUAN matches were found closer than 250 km. Thus, the AIRS comparison results, conducted at a variety of distance criteria, were used to illustrate the impact of the collocation criteria.

A few additional plots may be of interest. Figures 1 and 2 below show the impact of the distance coincidence criteria and mean percent difference between measurements of AIRS and GRUAN at 6 km and 8 km altitude, respectively. They show that the distance between the

measurements did not have a large impact on the reported comparison results, which used 500 km and 3 hour coincidence criteria.

[Figure]

Figure 1: Mean percent difference (black squares) and standard error in the mean (blue triangles) vs. distance between measurements for the AIRS − GRUAN comparison at 6 km altitude.

[Figure]

Figure 2: Same as Figure 1 but for 8 km altitude.

*The relevance of the respective satellite products is assessed with bulk departure statistics of satellite wrt reference data. The individual overall figures are then compared to the GCOS 5% requirements. Also important for climate is the ability to monitor the distribution of the moisture load and to track extreme situations. It would be informative to plot the retrieved (satellite) and reference (in situ, 125HR) WV distributions in the studied period and layers of interest, to confirm that dryest and moistest ends are correctly captured in the satellite products. In addition, a stratification of the statistics (bias/stddev...) in different WV-load bins would be instructive too in that perspective.*

The histograms of water vapour abundances from each instrument and at altitude levels of interest were plotted and compared. These show that the satellite instruments capture a similar distribution of water vapour abundances to those measured by the GRUAN-processed radiosondes. An example of this is shown in Figures 3, 4, and 5 below, which illustrate the water vapour abundances at 8 km from the GRUAN-processed radiosondes, AIRS, and ACE-FTS, respectively.

[Figure]

*Figure 3: Distribution of water vapour abundances above Eureka at 8 km altitude using the GRUAN-processed radiosonde data.*

[Figure]

*Figure 5: Distribution of water vapour abundances measured by AIRS at 8 km altitude within 250 km of Eureka.*

[Figure]

*Figure 4: Distribution of water vapour abundances measured by ACE-FTS at 8 km altitude within 500 km of Eureka.*

In Figure 4, AIRS appears to not capture the extreme wet (large) abundances captured by GRUAN, but the scale of the y-axis, the result of a very large number of measurements, hides the measurements of these values – which do indeed exist. In Figure 5, ACE-FTS does not capture some of the extreme wet (large) abundances captured by the AIRS and GRUAN datasets. However, ACE-FTS measurements are temporally limited to the spring and fall. Indeed, ACE-FTS measurements coincident with Eureka mostly occur in March. Figures 6, 7, and 8 illustrate the water vapour abundances at 8 km from GRUAN, AIRS, and ACE-FTS, but only include data acquired during March. This shows that the ACE-FTS measurements capture a similar distribution of water vapour abundances to the AIRS and GRUAN datasets.

[Figure]

*Figure 6: Distribution of water vapour abundances above Eureka at 8 km altitude using the GRUAN-processed radiosonde data acquired during March.*

[Figure]

*Figure 7: Distribution of water vapour abundances measured by AIRS at 8 km altitude within 250 km of Eureka, acquired during March.*

[Figure]

*Figure 8: Distribution of water vapour abundances measured by ACE-FTS at 8 km altitude within 500 km of Eureka, acquired during March.*

These figures were not included in the revision to the manuscript to avoid adding greater length. The reviewers have previously commented on the large number of figures in this work.

*Specific:*

*Page 6. Line 30-33: adding/Recalling figures illustrating MUSICA vertical sensitivity would be informative material to the reader.*

The following text has been added to refer the reader to specific figures that illustrate the MUSICA vertical sensitivity:

"The vertical sensitivity of the MUSICA retrieval is illustrated by Figures 2, 3, and 4 in Barthlott et al. (2017) and that for the MUSICA retrieval at the Eureka site is illustrated by Figure 4 in Weaver et al. (2017)."

*P14.L8: "but not THE information..." - missing word*
*P14.L8: typo/syntax "We use of the width..."*

These typos have been fixed.

*P19.L15: talking about "cloud-free" retrievals. AIRS products are generated also in cloudy scenes, using the cloud-clearing algorithm. Have such retrievals been included - cloud-clearing accounted for in the stratification/interpretation of the results?*

The AIRS data were filtered such that only the "best" quality level were used. This filtering results in the use of only cloud-cleared scenes.

*P20.L29-30: "the smoothing operation is not enough..." Isn't it also a possible sign that the DoFs is overestimated? I.e. that the assumed observation error in the OEM are underestimated for instance?*

This possibility has been added to the text, which now reads:

"Similarly, individual profile-to-profile comparisons with the nearest radiosonde profile show that TES profiles often capture the general shape of the lower tropospheric humidity profiles structure; however, the smoothing operation is not sufficient to bring the measurements into agreement. It is possible that the DOFS of the TES retrieval are overestimated."

*P21.L10: see second general comment*
*P21.L15-16: see third general comment*

See replies to general comments.

[revised manuscript text omitted]